# Mitochondrial complex III deficiency drives c-MYC overexpression and illicit cell cycle entry leading to senescence and segmental progeria

Janne Purhonen[1,2], Rishi Banerjee[1,2], Vilma Wanne[1,2], Nina Sipari [3], Matthias Mörgelin[4,5], Vineta Fellman [1,2,6,7] & Jukka Kallijärvi [1,2] ✉

Accumulating evidence suggests mitochondria as key modulators of normal and premature aging, yet whether primary oxidative phosphorylation (OXPHOS) deficiency can cause progeroid disease remains unclear. Here, we show that mice with severe isolated respiratory complex III (CIII) deficiency display nuclear DNA damage, cell cycle arrest, aberrant mitoses, and cellular senescence in the affected organs such as liver and kidney, and a systemic phenotype resembling juvenile-onset progeroid syndromes. Mechanistically, CIII deficiency triggers presymptomatic cancer-like c-MYC upregulation followed by excessive anabolic metabolism and illicit cell proliferation against lack of energy and biosynthetic precursors. Transgenic alternative oxidase dampens mitochondrial integrated stress response and the c-MYC induction, suppresses the illicit proliferation, and prevents juvenile lethality despite that canonical OXPHOS-linked functions remain uncorrected. Inhibition of c-MYC with the dominant-negative Omomyc protein relieves the DNA damage in CIII-deficient hepatocytes in vivo. Our results connect primary OXPHOS deficiency to genomic instability and progeroid pathogenesis and suggest that targeting c-MYC and aberrant cell proliferation may be therapeutic in mitochondrial diseases.

Substantial evidence from the past several decades suggests mitochondria as key modulators of aging[1]. Nevertheless, while mitochondrial diseases often shorten lifespan, premature aging per se is not a generally recognized feature of these diseases or their animal models. Moreover, most premature aging (progeroid) syndromes are due to compromised function of nuclear lamina proteins (e.g in Hutchinson-Gilford progeria syndrome, HGPS) or DNA repair enzymes (e.g. in Werner syndrome), both resulting in defective structural maintenance

and repair of the genome[2]. Intriguingly however, in yeast, increased nuclear genome instability in respiration-deficient mutant strains was reported already in the 1970s[3]. A later study found a similar response as a consequence of chemical inhibition of cellular respiration[4]. In mice, a mutation compromising the proofreading activity of the mitochondrial DNA (mtDNA) replicative polymerase γ leads to relatively late-onset premature aging-like manifestations[5,6]. These mice, coined mutator mice, accumulate mtDNA mutations, unavoidably

[1]Folkhälsan Research Center, Haartmaninkatu 8, 00290 Helsinki, Finland. [2]Stem Cells and Metabolism Research Program, Faculty of Medicine, University of Helsinki, P.O.Box 63, 00014 Helsinki, Finland. [3]Viikki Metabolomics Unit, University of Helsinki, P.O.Box 65, Helsinki, Finland. [4]Division of Infection Medicine, Department of Clinical Sciences, Lund University, P.O.Box 117, 221 00 Lund, Sweden. [5]Colzyx AB, Scheelevägen 2, 22381 Lund, Sweden. [6]Department of Clinical Sciences, Lund, Pediatrics, Lund University, P.O.Box 117, 221 00 Lund, Sweden. [7]Children's Hospital, Clinicum, University of Helsinki, P.O. Box 22, 00014 Helsinki, Finland. ✉e-mail: jukka.kallijarvi@helsinki.fi

leading to progressive loss of multiple mitochondrial functions. A recent study reported increased nuclear DNA damage response in the primary spermatocytes of and induced pluripotent stem cells derived from the mutator mice[7].

Nucleotide homeostasis is critical for cellular proliferation and genome maintenance[8–10]. This vulnerability is exploited in oncology, with several chemotherapeutic molecules targeting nucleotide biosynthesis in cancer cells. Mitochondria have a vital role in the biosynthesis of nucleotides and maintenance of their phosphorylation statuses[4,11–13]. Purine nucleotide biosynthesis connects to the respiratory electron transfer indirectly[11,13]. In contrast, pyrimidine biosynthesis depends directly on mitochondrial respiration because dihydroorotate dehydrogenase (DHODH) reduces ubiquinone (oxidized coenzyme Q) in order to oxidize dihydroorate to orotate[12]. Orotate then serves as a precursor for all pyrimidine nucleotides. Despite that animal cells lacking mtDNA are auxotrophic for pyrimidine nucleosides[14], it remains unknown whether DHODH function can become limiting in vivo in mitochondrial diseases. Mitochondrial respiration is vital also for the maintenance of the redox status of nicotinamide adenine dinucleotide (NAD$^+$ and NADH). Altered NAD(H) metabolism is a common feature of mitochondrial dysfunction and aging[15]. In cultured cells, mitochondrial defects can induce senescence via NAD$^+$/NADH imbalance[16]. Interestingly, several mouse models of mitochondrial disease show NAD$^+$ depletion[17,18]. Finally, the mitochondria-derived metabolites fumarate and succinate can act as oncometabolites that affect nuclear genome maintenance by inhibition of α-ketoglutarate-dependent dioxygenases including histone demethylases[19]. Taken together, mitochondrial dysfunction has the potential to cause cellular senescence and nuclear genome maintenance defects via multiple mechanisms. However, most findings from yeast and cultured mammalian cells remain uncorroborated in vivo.

GRACILE syndrome (Growth Restriction, Aminoaciduria, Cholestasis, Iron overload in the liver, Lactic acidosis, Early death) is a neonatally lethal mitochondrial disease caused by a homozygous missense mutation in *BCS1L (c.A232G, p.S78G)*, which encodes the translocase required for the incorporation of Rieske iron-sulfur protein (RISP, UQRCFS1) into respiratory complex III (CIII)[20]. The patients show liver and kidney disease but also unusual pre- and postnatal growth restriction, thin wrinkled skin, and lipodystrophy reminiscent of progeroid syndromes[20–22]. Mice bearing the analogous homozygous *Bcs1l[p.S78G]* knock-in mutation recapitulate the human syndrome apart from that they are born healthy and their CIII deficiency and the disease become evident only after 3 weeks of age[23].

Here, we utilize the *Bcs1l[p.S78G]* mice to ask if isolated CIII deficiency is sufficient to lead to cellular senescence and progeria-like disease. Our focused phenotyping led to the conclusion that severe CIII deficiency causes pathology strikingly similar to that of established progeroid syndromes. We identify widespread nuclear DNA damage response and cellular senescence in these mice, uncover mitochondria-dependent metabolic alterations affecting nuclear genome maintenance, and report interventions that remodel cellular proliferation to ameliorate the progeroid manifestations.

## Results

### CIII-deficient mice show cellular senescence and juvenile segmental progeria

A distinct subset of biological alterations, such as kyphosis, decreased bone mineral density and fat mass, alopecia, lymphoid depletion and thin skin, typically associated with advanced aging characterize progeroid syndromes in humans and mice[24]. We examined the *Bcs1l[p.S78G]* knock-in mice on two different mtDNA backgrounds for these alterations: *Bcs1l[p.S78G]* homozygotes on homoplasmic *mt-Cyb[p.D254N]* mtDNA background that have severe CIII deficiency and succumb by postnatal day (P) 35–40 with cachexia and hypoglycemia[25]; and the homozygotes

on wild-type (WT) mtDNA background that have similar juvenile disease onset and picture but slightly less severe CIII deficiency and survival past the juvenile metabolic crisis until deterioration due to late-onset dilating cardiomyopathy at approximately P200[26].

Quantification of the thoracic-lumbar curvature from radiographs of juvenile 1-month-old *Bcs1l[p.S78G];mt-Cyb[p.D254N]* mice confirmed severe kyphosis, as reported descriptively before[17,26,27] (Fig. 1a). Body composition analysis showed decreased body fat and bone mineral density in these (Fig. 1b, c) and adult mutant mice on WT mtDNA background (Supplementary Fig. 1a–c). GRACILE syndrome patients have thin wrinkled skin[21]. In *Bcs1l[p.S78G];mt-Cyb[p.D254N]* mice, skin histology showed absent dermal fat layer (Fig. 1d). Naturally-aged and *Zmpste24[-/-]* mice (modeling HGPS) have decreased excretion of major urinary proteins (MUPs) into urine[28,29]. *Bcs1l[p.S78G];mt-Cyb[p.D254N]* mice had albuminuria but greatly decreased urinary concentration of MUPs (Fig. 1e). The *Bcs1l[p.S78G]* mice had also repressed hepatic transcription of MUP genes and loss of urinary MUPs (Supplementary Fig. 1d, e). Similar to several progeroid mouse models, which often present involution of the lymphoid organs thymus and spleen[24], the thymic remnants of *Bcs1l[p.S78G];mt-Cyb[p.D254N]* mice typically weighted less than 10% of WT thymus (Fig. 1f, h). Their spleens were atrophic as well (Fig. 1g, h).

We next sought cellular and molecular signs of premature (non-replicative) senescence, a hallmark of premature aging, in the mutant mouse tissues. Previously, we reported an unusual hepatic progenitor cell response in *Bcs1l[p.S78G]* mice[27]. Normally, the liver regenerates via hypertrophy and proliferation of hepatocytes with little contribution from the progenitor cells unless hepatocyte proliferation fails[30,31]. Juvenile *Bcs1l[p.S78G]* and *Bcs1l[p.S78G];mt-Cyb[p.D254N]* mice, which should display both normal growth- and regeneration-associated hepatocyte proliferation, showed 10- to 30-fold increases in hepatic mRNA expression of the senescence marker CDKN1A (p21) (Fig. 1i), a cyclin-dependent kinase inhibitor blocking the cell cycle at G1 or G2[32]. The difference to WTs was at least as substantial at protein level (Fig. 1j). Another important effector of cellular senescence, the *p16[INK4A]* splice variant of *Cdkn2a*, the expression of which causes G1 arrest and occurs in Kupffer and liver endothelial cells but not in hepatocytes in aging or experimentally induced senescence[33,34], was not quantifiably expressed in P30 liver samples from any genotype (Supplementary Fig. 1f). Instead, the more abundant splice variant of *Cdkn2a* the *p19[ARF]*, which mediates G1 or G2 arrest[35], was induced in the mutant liver but more subtly than *Cdkn1a* (Supplementary Fig. 1f).

To confirm that cellular senescence follows from the CKDN1A-mediated cell cycle arrest, we quantified the aging pigment lipofuscin. It was undetectable in young (<P45) and restricted to rare sporadic cells in adult WT livers (Fig. 1k, l). In contrast, 1-month-old *Bcs1l[p.S78G];mt-Cyb[p.D254N]* mice had several fold higher lipofuscin levels than 7-month-old WTs. *Bcs1l[p.S78G]* mice, which live up to 7 months of age[26], developed massive lipofuscin accumulation after approximately 2.5 months of age. Based on β-galactosidase staining of liver cryosections, senescence-associated accumulation of lysosomes was also part of the senescence phenotype (Supplementary Fig. 1h). Furthermore, the mutant mice showed increased cleavage of histone H3 (Fig. 1m and Supplementary Fig. 1g), a chromatin modification linked to cellular senescence[36]. Cathepsin L, a lysosomal and nuclear cysteine protease implicated in nuclear lamina damage[37,38], performs this cleavage in cellular senescence[36]. Cathepsin L was transcriptionally upregulated in the mutant liver (Fig. 1n).

Further gene expression analyses focusing on the senescence-associated secretory phenotype (SASP) showed 50- to 600-fold induced amphiregulin (*Areg*) and increased expression of several chemokines and cytokines associated with SASP in the mutant liver (Supplementary Fig. 1i). Nevertheless, in line with the characterization of SASP in mitochondrial dysfunction[16], some typical SASP factors such as *Il6*, *Il15*, and *Cxcl15* were not induced at the whole tissue level. The most upregulated gene in the liver and kidney of adult *Bcs1l[p.S78G]* mice is

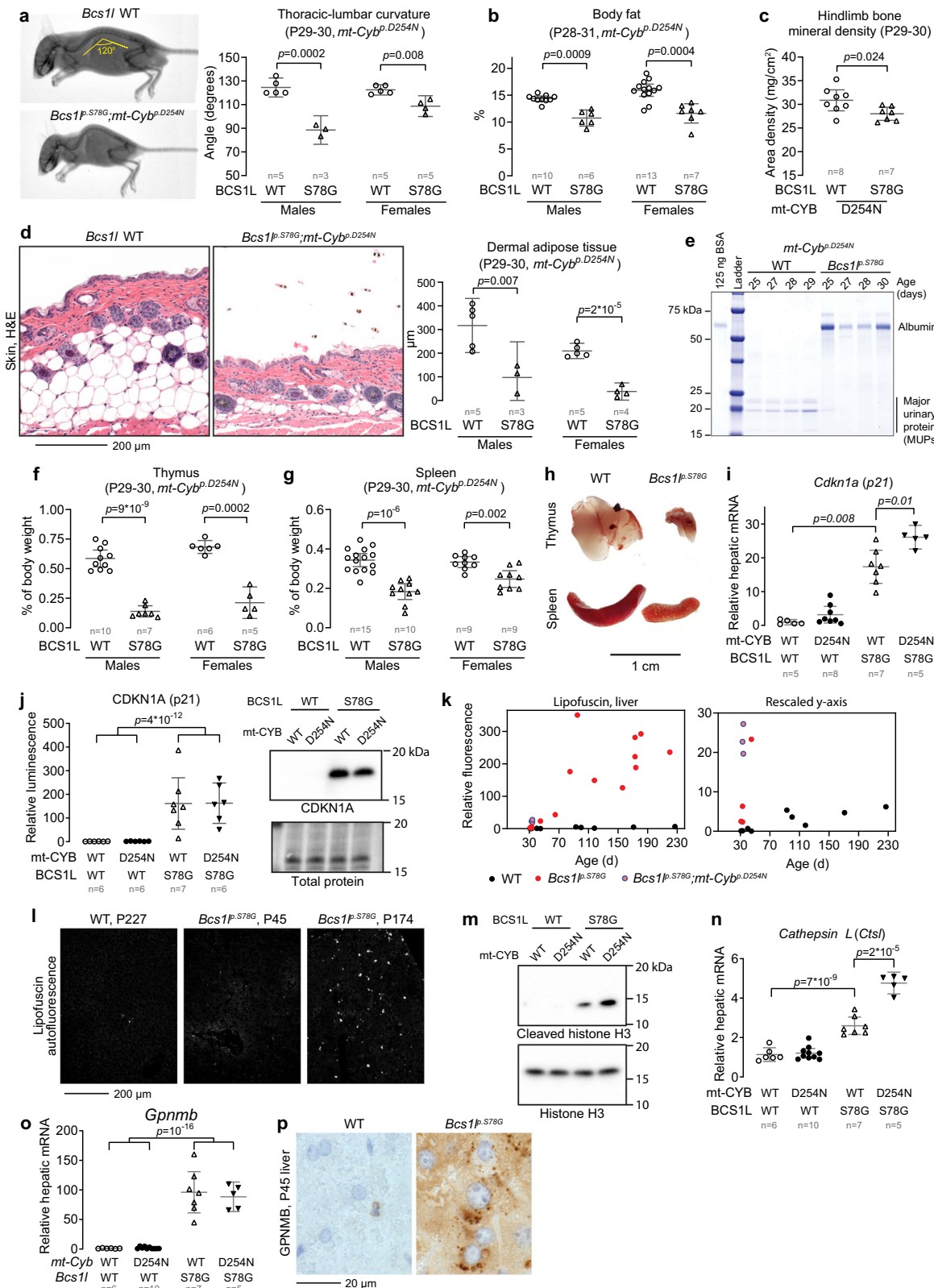

Gpnmb[26,27], an aging-associated transmembrane glycoprotein recently utilized as a means to target and eliminate senescent cells[39]. Here, we found that *Gpnmb* mRNA was increased 100-fold in the mutant mice already at one month of age (Fig. 1o). As a comparison, near-end-of-lifespan (27 months) WT mice have 35-fold increased hepatic levels of *Gpnmb* in comparison to young (4 months) mice (source data within ref. 39). GPNMB immunohistochemistry showed numerous

cytoplasmic foci in *Bcs1l^{p.S78G}* hepatocytes, while similar foci restricted to sporadic non-parenchymal cells in WT liver (Fig. 1p).

## Nuclear DNA damage in CIII-deficient tissues

Irreparable DNA damage commonly underlies cellular senescence[32]. To assess this possibility, we inspected transcriptomics data collected from adult (P150) mice of the *Bcs1l^{p.S78G}* strain[26]. Pathway analyses

**Fig. 1 | Progeroid phenotypes and cellular senescence in *Bcs1l^{p.S78G}* homozygous mice. a** DEXA scan of a *Bcs1l^{p.S78G};mt-Cyb^{p.D254N}* mouse and its healthy littermate illustrating the typical growth restriction and kyphosis of the mutants. The kyphosis was quantified from an angle formed by the thoracic and lumbar regions of the spine. **b** Echo-MRI quantification of body fat percentage. **c** DEXA quantification of hindlimb bone mineral density. **d** H&E-stained cross sections of skin and quantification of the dermal adipose tissue thickness. **e** SDS-PAGE analysis of male mouse urine for albuminuria and major urinary protein (MUP) (representative result of 4 similar experiments). **f** Weight of the thymus relative to body weight. **g** Weight of the spleen relative to body weight. **h** Representative images of thymi and spleens. **i** P30 hepatic gene expression of *Cdkna1*. **j** Western blot quantification and a representative blot of CDKN1A from P30 liver lysates. **k, l** Quantification and representative images of lipofuscin intrinsic fluorescence. **m** Representative Western blot of cleaved histone H3 in P30 liver lysates. Supplementary Fig. 1g shows a corresponding quantification. **n** *Ctsl* mRNA in P30 liver. **o** *Gpnmb* mRNA in P30 liver. **p** Immunostaining for GPNMB in liver sections. Abbreviations: WT wild type, P postnatal day. Statistics: Welch's two-sided *t*-test (**c**) or one-ANOVA followed by the selected pairwise comparisons with Welch's *t*-statistics. The error bars represent 95% CI of mean. All data points derive from independent mice. Source data are provided as a Source Data file.

suggested activated replication stress response in the liver and kidney of the *Bcs1l^{p.S78G}* mice (Fig. 2a). Fanconi anemia-related genes were also upregulated (Fig. 2b). Next, we compared our liver transcriptomics data against that of a well-characterized DNA repair-deficient progeroid mouse model (*Ercc1^{-/Δ7}* mice)[40]. Of the genes downregulated in the liver of *Ercc1^{-/Δ7}* mice, 68% were also downregulated in the *Bcs1l^{p.S78G}* livers (Supplementary Fig. 1j). Similar overlap for upregulated genes was 41%. Then, we assessed the DNA double-strand break marker γH2AX (S139-phophorylated histone H2AX). Its amount in liver lysates from the mutant mice was increased several fold (Fig. 2c). Immunohistochemistry showed that the hepatic γH2AX signal derived exclusively from hepatocytes (Fig. 2d, e). The γH2AX staining pattern in the hepatocytes of young CIII-deficient mice ranged from nuclear foci (active DNA damage) to prominent pan-nuclear staining, similar to described for lethal replications stress[41]. Increased number of 53BP1 foci in *Bcs1l^{p.S78G};mt-Cyb^{p.D254N}* hepatocyte nuclei further confirmed the DNA damage (Fig. 2f). In 7-month-old *Bcs1l^{p.S78G}* mice, the pan-nuclear γH2AX staining pattern was less prominent than in the younger counterparts but, overall, γH2AX-positive hepatocytes were more prevalent (Supplementary Fig. 1k).

Renal cortex tubules, which are severely affected by the *Bcs1l^{p.S78G}* mutation in humans and mice, also showed a high number of γH2AX-positive nuclei in the mutant tissue (Fig. 2e). In skeletal muscle, a mostly post-mitotic tissue, γH2AX amount was low and not significantly different across the genotypes (Supplementary Fig. 2a, b). The mutant bone marrow and small intestine, two tissues with a high-rate of stem cell-mediated renewal, did not show elevated γH2AX levels (Fig. 2g and Supplementary Fig. 2c). Finally, the exocrine pancreas, which regenerates via replication of differentiated cells similar to the liver and renal cortex tubules[42,43], of mutant mice contained an abnormally high number of γH2AX-positive nuclei (Fig. 2h).

## CIII deficiency leads to laminopathy-like nuclear alterations
Aberrant nuclear architecture can be both the cause and the consequence of DNA damage, and it is also a hallmark of cellular senescence and progeroid syndromes[32,44]. *Bcs1l^{p.S78G}* livers showed hepatocyte anisokaryosis and karyomegaly (Fig. 3a–d), suggesting cell cycle arrest at G2 phase and atypical cell cycle-related increase in polyploidy[45–47]. Essentially all WT hepatocyte nuclei were evenly round in shape, whereas mutant hepatocytes had frequent laminopathy-like nuclear envelope invaginations and blebs (Fig. 3a–c, e). After approximately P100, some *Bcs1l^{p.S78G}* mutant hepatocyte nuclei contained cytoplasmic inclusion-like structures, which were very similar to those described in the hepatocytes of DNA repair-deficient *Xpg^{-/-}* and *Ercc^{-/Δ}* mice and also in normal advanced liver aging[45,48,49].

Immunostaining for lamin A/C (LMNA) in liver sections from juvenile *Bcs1l^{p.S78G}* mice revealed a fraction of hepatocytes with a diffuse nucleoplasmic LMNA distribution instead of the normal rim-like pattern beside the nuclear envelope (Supplementary Fig. 2e). In older *Bcs1l^{p.S78G}* mice, the staining reflected the nuclear structural abnormalities as described above. Western blot of liver lysates from juvenile *Bcs1l^{p.S78G}* mutants showed an altered ratio of lamin C to lamin A (Supplementary Fig. 2f). qPCR revealed this change to be of transcriptional origin (Fig. 3f). This shift in the splicing of *Lmna*

prompted us to quantify *progerin*, a pathogenic transcript resulting from the use of a cryptic *lamin A* splice site[50]. *Progerin* expression has thus far been reported in HGPS patients with the *C1824T* splice site mutation and in mice with the analogous mutation, and as an extremely infrequent transcript in normal cells[50]. The *Bcs1l^{p.S78G}* mice showed significant *progerin* expression in the liver (Fig. 3g and Supplementary Fig. 2g).

## Bypass of the CIII-CIV segment prevents the DNA damage and juvenile lethality
To interrogate which consequences of the CIII deficiency might underlie the nuclear DNA damage and senescence, we utilized transgenic expression of *Ciona intestinalis* alternative oxidase (AOX)[26] in the *mt-Cyb^{p.D254N}* mtDNA background to restore coenzyme Q oxidation and electron flow upstream of CIII and complex IV (CIV).

AOX expression tripled the median survival of *Bcs1l^{p.S78G};mt-Cyb^{p.D254N}* mice to 111 days with some mice living more than 200 days (Fig. 4a). AOX prevented the weight loss and failure to grow (Fig. 4b), as well as the typical liver pathology (Fig. 4c and Supplementary Fig. 3), and essentially preserved the function and morphology of the kidney (Fig. 4d, e and Supplementary Fig. 4). At molecular level, AOX abolished the DNA damage response and cellular senescence in the liver based on the lack of γH2AX, TP53 (p53), and CDKN1A induction (Fig. 4f–h). Moreover, AOX completely prevented the hepatic expression of the senescence marker GPNMB (Fig. 4i). Interestingly, in the kidney, AOX partially prevented the induction of γH2AX but not CDKN1A or GPNMB (Fig. 4j–l), suggesting attenuated active DNA damage but not cell cycle arrest in this tissue. AOX also preserved near-normal *Lmna* splicing, including the absence of *progerin* expression, and prevented the hepatocyte nuclear envelope aberrations (Fig. 4m–o).

Similar to in *Bcs1l^{p.S78G};mt-Cyb^{p.D254N}* mice, AOX also prevented the hepatic induction of γH2AX and CDKN1A, and attenuated the liver disease in the juvenile (P35) *Bcs1l^{p.S78G}* mice (Supplementary Fig. 5a, c–e). However, by P150, the beneficial effects of AOX had faded (Supplementary Fig. 5b and ref. 26).

## The beneficial effect of AOX does not depend on canonical OXPHOS-linked functions
Inefficient respiratory electron transfer and mitochondrial ATP production are considered main disease mechanisms in OXPHOS diseases. In isolated CIII-deficient liver mitochondria, AOX improved the respiratory electron transfer to some degree (Fig. 5a, b), while the effect was opposite in the kidney mitochondria (Supplementary Fig. 6a–c). Peculiarly, the AOX inhibitor propyl gallate almost completely blocked the respiration of liver mitochondria from AOX-expressing *Bcs1l^{p.S78G};mt-Cyb^{p.D254N}* mice (Fig. 5c), implicating that AOX did not operate in parallel with CIII but instead had taken over its residual function. Analysis of CIII composition verified this conclusion by showing further exacerbation of the CIII assembly defect upon AOX expression (Fig. 5d, e and Supplementary Fig. 6d, e). In line with this, the CIII-deficient liver mitochondria showed decreased ATP production and were unable to maintain membrane potential with NADH-generating substrates under saturating ADP concentration in the

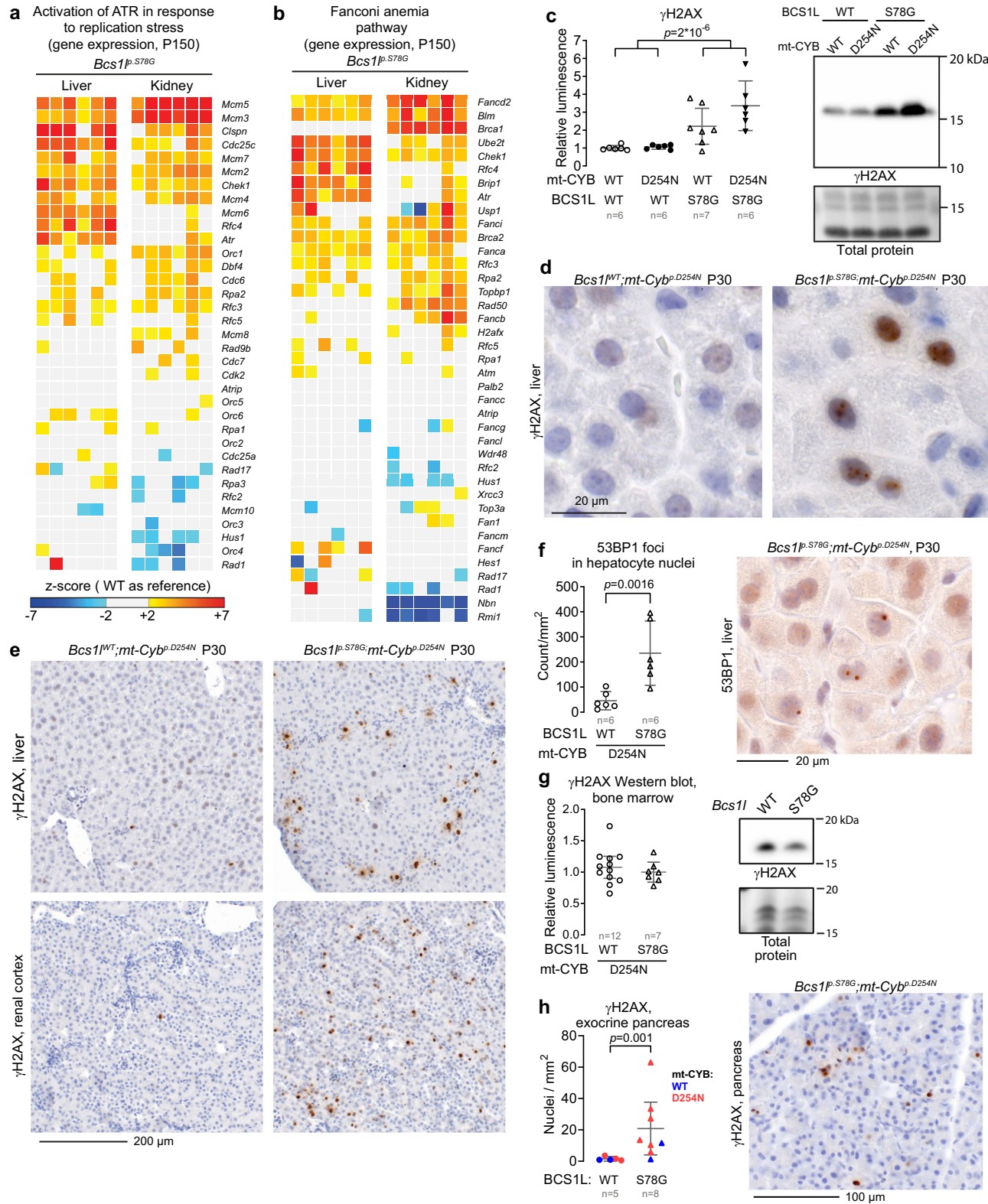

**Fig. 2 | Nuclear DNA damage in CIII-deficient mice. a, b** Heat map visualization of liver and kidney transcriptomics data overlapping with "activation of ATR in response to replication stress" (**a**) and "Fanconi anemia" (**b**) pathways (Reactome database). **c** Western blot quantification and representative blot of γH2AX from P30 liver lysates as a marker for DNA double-strand breaks. **d, e** γH2AX immunohistochemistry showing representative staining patterns in hepatocyte nuclei (**d**) and the typical frequency of γH2AX-positive nuclei in the liver and renal cortex (**e**). The selection of representative micrographs was based on analysis of more than 10 mice/genotype. **f** Number of 53BP1 foci in hepatocyte nuclei as a marker for activation of DNA repair by nonhomologous end-joining. **g** Western blot quantification of γH2AX from P30 bone marrow lysates. **h** Quantification of γH2AX-positive nuclei in P32-36 exocrine pancreas and a representative immunostaining. Statistics: one-way ANOVA followed by the selected pairwise comparisons (Welch's *t*-statistics) or Welch's two-sided *t*-test (**f, h**). The error bars represent 95% CI of mean. All data points derive from independent mice. Source data are provided as a Source Data file.

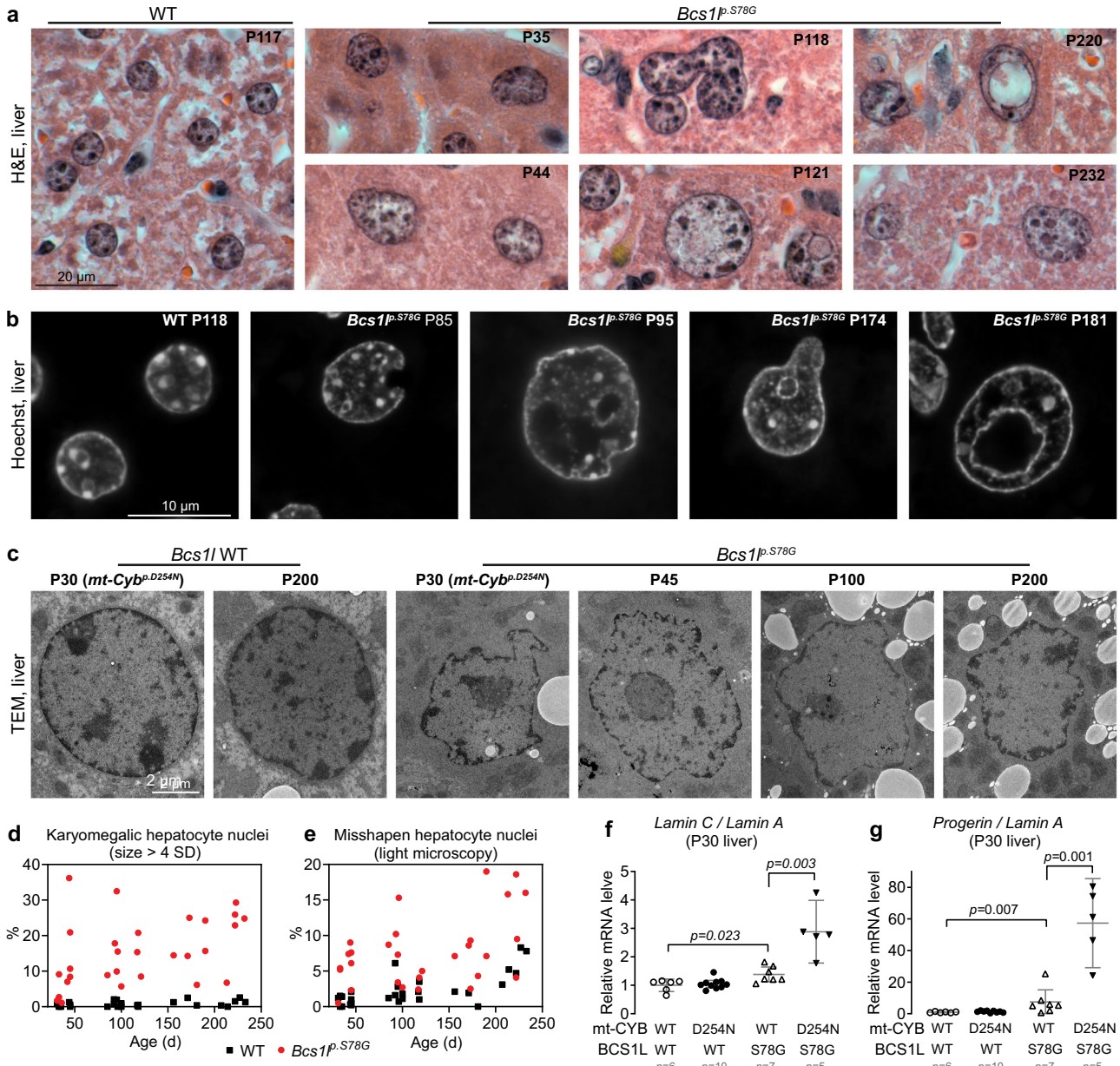

**Fig. 3 | Morphological aberrations of CIII-deficient hepatocyte nuclei. a** H&E-stained liver sections showing typical WT hepatocyte nuclei and the spectrum of aberrant *Bcs1l*[p.S78G] hepatocyte nuclei (anisokaryosis, karyomegaly, uneven nuclear envelope, atypical shapes, nuclear envelope blebs and invaginations, intranuclear cytoplasmic inclusion-like structures). **b** High-magnification images of Hoechst-stained hepatocyte nuclei. **c** Representative electron micrographs of hepatocyte nuclei. Two samples from the *mt-Cyb*[p.D254N] background are indicated in parentheses. Supplementary Fig. 2d shows the corresponding whole-cell cross sections. **d** Quantification of karyomegalic hepatocyte nuclei defined as cross-sectional area greater than 4 SDs of normal (WT hepatocytes as reference). **e** Quantification of misshapen hepatocyte nuclei based on nuclear envelope blebs, invaginations, and inclusion bodies from H&E-stained sections. **f, g** qPCR analysis of *Lmna* splice variants. Statistics: one-way ANOVA followed by the selected pairwise comparisons (Welch's *t*-statistics). The error bars represent 95% CI of mean. All data points derive from independent mice. Source data are provided as a Source Data file.

presence of AOX (Fig. 5f, g). At tissue level, AOX did not significantly alter hepatic ATP levels (Fig. 5h).

In certain cultured celllines, the rescue of NAD(H) redox balance is sufficient to prevent mitochondrial dysfunction-associated cellular senescence[16]. Unexpectedly, in contrast to some in vitro studies[51,52], AOX did not correct the elevated hepatic NADH-to-NAD+ ratio in *Bcs1l*[p.S78G];*mt-Cyb*[p.D254N] mice (Fig. 5i). Accumulation of succinate and fumarate, two mitochondria-derived oncometabolites, can inhibit homologous recombination, a branch of DNA double-strand break repair[19]. Some *Bcs1l*[p.S78G];*mt-Cyb*[p.D254N] mice showed increased hepatic succinate and fumarate levels (Fig. 5j, k). However, AOX appeared to only exacerbate their accumulation.

In conclusion, the CIII-CIV segment proved entirely dispensable for normal liver function in the presence of AOX despite the prevailing canonical detrimental consequences of OXPHOS dysfunction.

## Mitochondrial ROS do not explain the nuclear DNA damage and senescence

Mitochondria produce reactive oxygen species (ROS), which can potentially cause oxidative damage to mitochondrial and nuclear DNA. The predicted effect of *BCS1L* mutations is decreased superoxide production at CIII due to the loss of the vital catalytic RISP subunit, the loss of which prevents the formation of an unstable semiquinone in the quinol oxidation site of CIII and thus electron leak to molecular

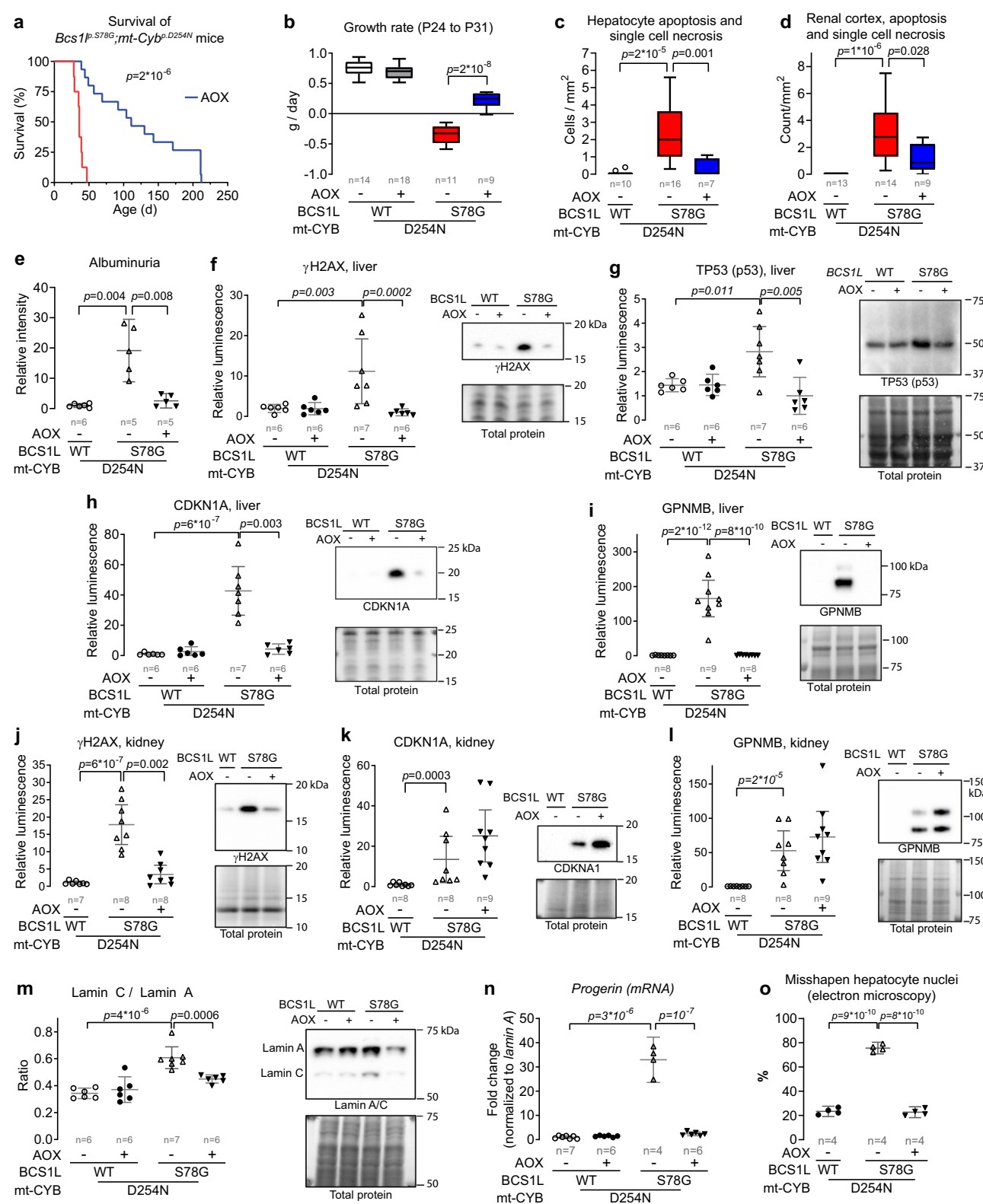

oxygen[53]. However, CIII deficiency may cause reverse-electron flow and increased superoxide production at CI[54]. To test these scenarios, we measured absolute mitochondrial ROS production after removal of bias by endogenous antioxidant systems[55]. Liver mitochondria from *Bcs1l* WT and *Bcs1l^{p.S78G};mt-Cyb^{p.D254N}* mice respiring with NADH-generating substrates had similar $H_2O_2$ production (Fig. 5l). To induce reverse-electron flow, we added a high concentration of succinate with or without NADH-generating substrates into the assay. The

mitochondria from *Bcs1l^{p.S78G};mt-Cyb^{p.D254N}* mice had lower $H_2O_2$ production than from *Bcs1l* WTs in the presence of succinate (Fig. 5l and Supplementary Fig. 6f). AOX further decreased the $H_2O_2$ production under high succinate concentration. Finally, as a read out for cellular ROS damage, we measured the redox statuses of a mitochondrial and a cytosolic peroxiredoxin in the liver. The mitochondrial peroxiredoxin (PRDX3) was less oxidized in the mutants than in *Bcs1l* WTs (Fig. 5m). In contrast, the redox status of the cytosolic counterpart, PRDX1, was

**Fig. 4 | AOX-mediated bypass of the CIII-CIV segment abolishes the DNA damage response and prevents the juvenile lethality of *Bcs1l*[p.S78G];*mt-Cyb*[p.D254N] mice. a** Survival analysis of *Bcs1l*[p.S78G];*mt-Cyb*[p.D254N] mice without (*n* = 8) and with AOX expression (*n* = 15). **b** Growth rate of the mice normalized for sex (estimated marginal means and residuals from a linear model). **c, d** Quantification of apoptotic and necrotic cell remnants from H&E-stained liver and kidney sections. **e** SDS-PAGE analysis of albuminuria as a measure of kidney function (Supplementary Fig. 4d shows a representative gel). **f–i** The effect of AOX on the hepatic expression of γH2AX, TP53, CDKN1A, and GPNMB. For the Western blot detection of γH2AX and CDKN1A, the liver lysates were run on SDS-PAGE with Tris-Tricine buffers, which slightly affects the migration of CDKN1A in comparison to Tris-glycine buffering system used elsewhere in this study. **j–l** The Effect of AOX on γH2AX, CDKN1A and GPNMB levels in kidney lysates. Mostly renal cortex was sampled for the γH2AX and CDKN1A Western blots, whereas whole kidney lysates (cortex + medulla) were used for GPNMB quantification. **m** Western blot quantification of lamin C-to-lamin A ratio in liver lysates and a representative blot. **n** The effect of AOX on hepatic *progerin* expression (mRNA). **o** Quantification of early-onset hepatocyte nuclear aberrations by electron microscopy. All data are from P30 mice unless otherwise stated in the figures. Statistics: Log-rank test for survival analysis and one-way ANOVA followed by the selected pairwise comparisons (Welch's *t*-statistics) for other data. The error bars represent 95% CI of mean. The box blots show the median, the quartiles, and the minimum and maximum. All data points derive from independent mice. Source data are provided as a Source Data file.

similar across the genotypes (Fig. 5n). Collectively, our data do not support mitochondrial ROS as a cause of the DNA damage and senescence in *Bcs1l*[p.S78G] mice.

## CIII deficiency amplifies hepatic biosynthesis of pyrimidine nucleosides

De novo pyrimidine nucleoside biosynthesis depends on CIII via ubiquinone, which functions as the electron acceptor for the mitochondrial DHODH enzyme[12]. In cultured cells, the blockade of DHODH or CIII can trigger p53 activation, replication stress, and cell cycle arrest due to compromised pyrimidine biosynthesis[52,56,57]. AOX expression can restore the growth of respiration-deficient cancer cells by allowing DHODH to function independently of CIII[52,57]. The livers of *Bcs1l*[p.S78G];*mt-Cyb*[p.D254N] mice accumulated aspartate, carbamoyl-aspartate and dihydroorotate, three metabolites upstream of DHODH (Fig. 6a). However, orotate, the reaction product, also accumulated (Fig. 6a), suggesting increased flux through the pathway or blockade downstream of DHODH. AOX attenuated the accumulation of carbamoyl-aspartate, dihydroorotate, and orotate, but did not affect aspartate levels. To assess DHODH function more directly, we measured the conversion of dihydroorotate to orotate in isolated liver mitochondria. The maximal ubiquinone-dependent DHODH activity was increased in comparison to WTs in *Bcs1l*[p.S78G];*mt-Cyb*[p.D254N] and even more so in *Bcs1l*[p.S78G] mice (Fig. 6b). These differences were not due to altered expression of DHODH (Supplementary Fig. 7a, b). When DHODH had to compete with CI for the availability of ubiquinone, the orotate production in *Bcs1l*[p.S78G];*mt-Cyb*[p.D254N] mitochondria decreased to WT levels, while in *Bcs1l*[p.S78G] mitochondria it still remained elevated (Fig. 6b). Surprisingly, AOX did not improve the DHODH function. Finally, we supplemented the chow of *Bcs1l*[p.S78G];*mt-Cyb*[p.D254N] mice with orotate to bypass DHODH. The orotate supplementation did not attenuate hepatic H2AX phosphorylation or CDKN1A expression (Fig. 6c, d). In accordance with a recent publication showing that fumarate reduction-driven ubiquinol oxidation can maintain DHODH activity in mouse liver and several other tissues[58], our data strongly suggest that, unlike in cancer cell models, DHODH function was not limiting and not responsible for the DNA damage and cellular senescence in vivo. Rather, the most plausible interpretation of our data is that pyrimidine nucleotide biosynthesis was amplified.

## CIII deficiency results in severe nucleotide depletion and imbalance

Considering the importance of mitochondria in the biosynthesis of not only pyrimidine but all nucleotides and in the maintenance of their redox and phosphorylation statuses, we quantified ribonucleoside triphosphates (rNTPs), the main pyridine dinucleotides, and deoxyribonucleosides triphoshptes (dNTPs) in the liver of P30 mice. *Bcs1l*[p.S78G];*mt-Cyb*[p.D254N] mice showed depletion of all rNTPs (Fig. 6e). AOX did not correct this. Uracil accumulation suggested increased RNA catabolism for nucleotide salvage or defective uracil ribosylation and phosphorylation into nucleotides (Fig. 6f). Measurement of total RNA and 18s rRNA confirmed the hepatic ribosomal RNA depletion (Fig. 6g, h). Nevertheless, labeling of nascent liver and kidney RNA indicated increased transcription (Fig. 6i, j). This together with the rNTP and RNA depletion strongly suggested transcription stress and excessive anabolism in the mutant liver.

*Bcs1l*[p.S78G];*mt-Cyb*[p.D254N] mice also showed hepatic depletion of the main pyridine dinucleotides, NAD(H) and NADP(H) (Fig. 6e). Despite the lack of effect on NAD(H) redox status (Fig. 5j), AOX increased the total pool sizes of NAD(H) and NADP(H) (Fig. 6e). However, our previous study[17] showed that nicotinamide riboside-based NAD(H) repletion had no effect on liver disease progression in the *Bcs1l*[p.S78G];*mt-Cyb*[p.D254N] mice, suggesting that the effect of AOX on NAD(H) pool size was not instrumental to the prevention of tissue pathology.

Despite the rNTP depletion, dNTPs, which derive from ribonucleotides, were not depleted in the mutant liver tissue (Fig. 6e), perhaps because their levels are under tight allosteric feedback regulation[59]. Instead, the concentrations of purine dNTPs (dATP and dGTP) were increased, indicating a skewed balance or altered proliferative status of the liver, or both. In contrast, in AOX-expressing mutant mice, hepatic dATP, dCTP and dTTP were decreased below WT level (Fig. 6e). In addition to being needed for the replication of the genome during the cell cycle, balanced levels of dNTPs are vital for mtDNA maintenance. However, the skewed hepatic dNTP pools did not cause mtDNA depletion (Supplementary Fig. 7c, d), which is in line with normal enzymatic activities of CI and CIV in the mitochondria of *Bcs1l*[p.S78G] homozygotes[25,26]. Finally, because increased mtDNA turnover has been suggested to cause dNTP imbalance[7], we assessed the rate of mtDNA synthesis by measuring in vivo bromodeoxyuridine (BrdU) incorporation into mtDNA. *Bcs1l*[p.S78G];*mt-Cyb*[p.D254N] mice did not show increased hepatic mtDNA synthesis (Supplementary Fig. 7e).

In summary, *Bcs1l*[p.S78G];*mt-Cyb*[p.D254N] mice showed drastic depletion and imbalance of nucleotides, which is a well-established cause of transcription stress and cell cycle arrest, but also of replication stress and DNA damage upon G1-checkpoint evasion and cell cycle progression to the S-phase[8–10].

## AOX dampens mt-ISR and blunts cancer-like c-MYC induction

A clue to the puzzling beneficial effect of AOX came from liver gene expression analyses, which showed that AOX attenuated the *Bcs1l*[p.S78G]-induced upregulation and downregulation of proliferation-associated genes controlling nucleotide biosynthesis and catabolism, respectively (Fig. 6k). AOX also attenuated the remodeling of serine-glycine-one-carbon metabolism according to metabolite changes related to this important pathway in nucleotide biosynthesis[60] (Fig. 6l). These processes are known targets of the mitochondrial integrated stress response (mt-ISR)[61,62]. Further analysis of mt-ISR markers indicated a robust suppression of mt-ISR by AOX (Fig. 7a). Nevertheless, AOX did not suppress all stress signals as the expression of *Gdf15*, a mitochondrial dysfunction-associated mitokine, remained elevated (Fig. 7b). The upstream branches of the integrated stress response converge on the phosphorylation of eukaryotic initiation factor 2 alpha (eIF2α) at serine 51[63]. In the liver of *Bcs1l*[p.S78G];*mt-Cyb*[p.D254N] mice, eIF2-α

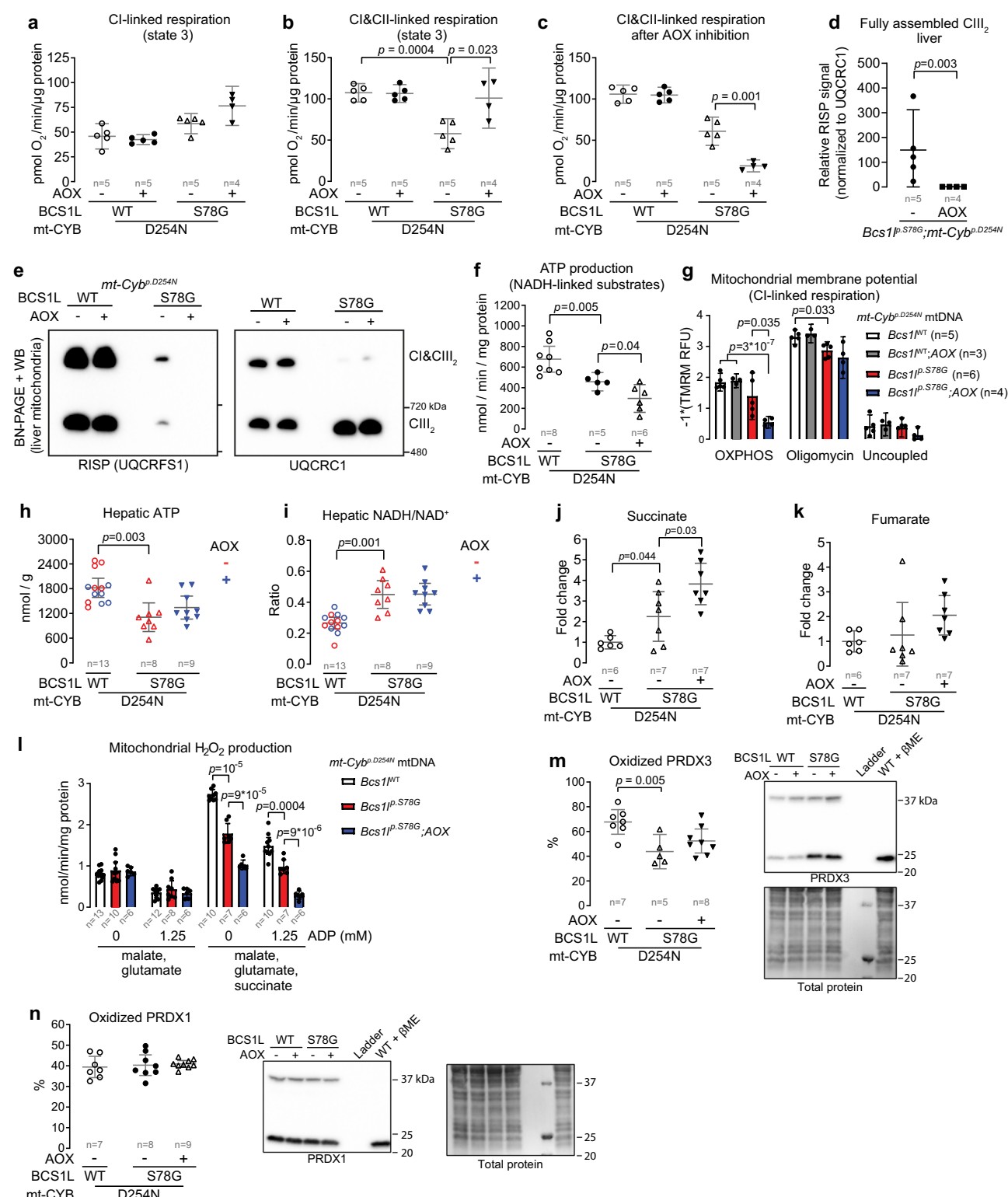

phosphorylation was increased roughly 50-fold (Fig. 7c). AOX markedly blunted this response. A key transcriptional mediator of mt-ISR is ATF4[63]. This transcription factor was highly increased in the mutant liver nuclei, but not in the presence of AOX (Fig. 7d).

Hyperactivity of the mTORC1 pathway accompanies mt-ISR in some disease settings[61]. RPS6 phosphorylation status as a readout, $Bcs1l^{p.S78G};mt\text{-}Cyb^{p.D254N}$ livers had a tendency for increased mTORC1 activity, but this was unaffected by AOX (Fig. 7e), implying that mTORC1 was not a key player in the mt-ISR of CIII deficiency.

In several published transcriptomics datasets, c-MYC upregulation accompanies mt-ISR, albeit with entirely unknown significance[51,62,64,65]. In our transcriptomics data from P45 mutant liver, c-MYC was the top predicted transcriptional regulator explaining the overall gene expression changes[27]. In the P30 liver of $Bcs1l^{p.S78G};mt\text{-}Cyb^{p.D254N}$ mice, $Myc$ mRNA levels were increased 40-fold (Fig. 7f). AOX almost totally abolished this upregulation. The differences were of similar magnitude at protein level (Fig. 7g). $Myc$ mRNA expression was elevated also in the $Bcs1l^{p.S78G}$ mice in the liver, kidney, and heart

**Fig. 5 | Effect of AOX on OXPHOS, NAD(H) redox status, and mitochondrial ROS production in Bcs1l[p.S78G];mt-Cyb[p.D254N] livers. a** Phosphorylating respiration in liver mitochondria in the presence of malate, glutamate, pyruvate, and ADP (CI-linked respiration). **b** Phosphorylating respiration of liver mitochondria in the presence of malate, glutamate, pyruvate, succinate, and ADP (CI&CII-linked respiration). **c** CI&CII-linked respiration after inhibition of AOX with 50 μM propyl gallate. **d**, **e** Blue-native PAGE analysis of CIII assembly from digitonin-solubilized liver mitochondria. The RISP subunit is present only in the fully assembled CIII dimers. **f** ATP production in liver mitochondria during CI-linked respiration. **g** Membrane potential of liver mitochondria during CI-linked phosphorylating respiration (OXPHOS). The subsequent addition of ATP-synthase inhibitor, oligomycin A, and the uncoupler, SF-6847, were used to obtain maximal and dissipated membrane potential, respectively. **h** Enzymatic determination of ATP in liver tissue. **i** Hepatic NADH-to-NAD$^+$ ratio. **j**, **k** Succinate and fumarate levels in liver. **l** Measurement of H$_2$O$_2$ production in isolated liver mitochondria. The mitochondrial antioxidant systems and carboxylesterases were blocked as described in the Materials and Methods. **m**, **n** Non-reducing Western blot of PRDX3 (**m**) and PRDX1 (**n**). One sample treated with β-mercaptoethanol (βME) served as a control to verify the specificity of bands corresponding to peroxiredoxin dimers. All data are from 1-month-old mice. Statistics: **d**, $\chi^2$-test on binarized data (detected or not detected); all other data, one-way ANOVA followed by the selected pairwise comparisons (Welch's $t$-statistics). The error bars represent 95% CI of mean. All data points derive from independent mice. Source data are provided as a Source Data file.

(Fig. 7h), suggesting a systemic response to the CIII deficiency. In addition to the liver, AOX also blunted *Myc* upregulation in the skeletal muscle and kidney (Fig. 7i, j).

## c-MYC upregulation precedes the DNA damage and cell cycle arrest

Extensive literature show that excessive c-MYC drives anabolism and cellular proliferation at all cost, even against lack of resources, a scenario leading to oncogene-induced replication stress and genomic instability[66]. Therefore, we asked whether the apparently exaggerated c-MYC upregulation precedes the DNA damage response and cell cycle arrest in the CIII-deficient mice. Indeed, *Bcs1l[p.S78G];mt-Cyb[p.D254N]* mice had elevated hepatic c-MYC expression already immediately after weaning (P18-P25, presymptomatic period), whereas γH2AX and CDKN1A were induced only after P26 (Fig. 8a). AMP-activated protein kinase (AMPK), a cellular sensor of ATP and glucose availability, can counterbalance c-MYC-driven rewiring of metabolism towards growth and proliferation[67]. Unexpectedly, AMPK activity, based on its activating phosphorylation, linearly decreased from weaning to the age one month in the liver irrespective of the *Bcs1l[p.S78G]* mutation (Fig. 8a). While the phosphorylation status was marginally increased (Fig. 8a and ref. 17), the mutant livers showed diminished levels of the AMPK α-subunit (Fig. 8b, c), suggesting insufficient sensing of the cellular energy status by AMPK to counterbalance c-MYC. Unexpectedly as well, the canonical mt-ISR based on eIF2α phosphorylation did not precede but followed the c-MYC upregulation (Fig. 8a), suggesting that mt-ISR is downstream of c-MYC, analogous to c-MYC-driven excessive anabolic metabolism leading to integrated stress response in cancer cells[68].

Intriguingly, we previously reported that essentially complete withdrawal of dietary carbohydrates in the form of ketogenic diet, a promising approach to limit cancerous proliferation in preclinical studies[69], attenuates the liver disease of *Bcs1l[p.S78G]* mice and decreases the senescence markers lipofuscin and *Gpnmb* expression[27]. Here, we found that the carbohydrate restriction suppressed the *Myc* upregulation and γH2AX induction in the mutant mice (Supplementary Fig. 8a, b), likely explaining its beneficial effect.

A common feature of c-MYC-driven cell cycle progression in cancer is bypass of metabolic checkpoints[66]. This happens also in normal cells[70]. Therefore, we assessed a series of cell cycle markers (Fig. 9a and Supplementary Fig. 9a). Unexpectedly, given their overwhelming nucleotide depletion, which should suppress cell cycle entry or progression to the S-phase[9], but in line with the copious c-MYC upregulation, *Bcs1l[p.S78G];mt-Cyb[p.D254N]* mice had elevated hepatic expression of cyclin D1 and PCNA, the expression of which peaks at G1- and S-phase of the cell cycle, respectively, suggesting inadequate metabolic checkpoints at G0 and G1. In contrast, cyclin A2 and Ki-67 levels, the expression of which peaks at G2 to early mitosis, were not increased. Stunningly, AOX expression plunged these four proliferation markers to below WT level, implying that limiting cell cycle entry was key to the rescue of tissue pathology. In line with the decreased proliferation markers, AOX expression prevented the accumulation of large and atypically large hepatocyte nuclei, in other words, cells in S-G2 phase of the cell cycle and cells with increased polyploidy (Fig. 9b, c). In fact, the proportion of large nuclei upon AOX expression was less than in normal age-matched liver.

## Illicit proliferation, replication stress and mitotic catastrophes link CIII deficiency to genomic instability

To confirm the illicit cell cycle progression to the S-phase, we exposed the mice to a 16-h in vivo BrdU labeling of replicating DNA. *Bcs1l[p.S78G];mt-Cyb[p.D254N]* mice showed marked increase in hepatic and renal DNA synthesis (Fig. 9d, f and Supplementary Fig. 9b, e). Despite this, their liver sections were surprisingly devoid of mitotic figures, normally readily observed in growing young mice (Fig. 9h). The mitotic figures were also missing in the liver sections of AOX-expressing *Bcs1l[p.S78G];mt-Cyb[p.D254N]* mice. In *Bcs1l* WTs, the BrdU-labeling index and the number of mitotic cells (histone H3 Ser10 phosphorylation) showed a strong correlation in both liver and kidney (Fig. 9d–g and Supplementary Fig. 9b–e). In *Bcs1l[p.S78G];mt-Cyb[p.D254N]* mice, the number of cells in mitosis was minimal, and these two markers of proliferating cells showed no correlation. Thus, most *Bcs1l[p.S78G];mt-Cyb[p.D254N]* hepatocytes and renal tubular epithelial cells failed to complete the cell cycle and became arrested in the G2 phase, resulting in polyploidy or aneuploidy, or both, and senescence or cell death.

To pinpoint the CIII deficiency-induced DNA damage to the proliferating cells, we quantified γH2AX foci in cyclin A2-positive hepatocytes. While hepatocytes with strong pan-nuclear γH2AX staining, potentially a pre-apoptotic response to lethal replication stress[41], were generally not cyclin A2 positive, the *Bcs1l[p.S78G];mt-Cyb[p.D254N]* hepatocytes positive for cyclin A2 showed increased number of γH2AX foci (Fig. 9i, j). To assess whether cellular proliferation is a prerequisite for DNA damage upon CIII inhibition, we treated AML12 cells, a mouse hepatocyte line forced to proliferate by transgenic expression of human *TGFA*, with the CIII inhibitors myxothiazol and antimycin A and assessed γH2AX as a proxy for DNA damage. Both inhibitors strongly induced γH2AX in proliferating but not in quiescent cells (Fig. 9k and Supplementary Fig. 9f).

Finally, we assessed the morphology of mitotic cells, the abnormalities of which typically arise from replications stress[71–73], from liver sections of P45 *Bcs1l[p.S78G]* mice. We found frequent abnormalities such as multipolar mitotic spindles and anaphases, lagging or dispersed chromatin, and anaphase bridges (Fig. 9l, m and Supplementary Fig. 9g), abnormalities strongly linked to genomic instability[73].

## Functional blockade of c-MYC alleviates the DNA damage in the liver

To obtain causal evidence for the role of c-MYC as a driver of the DNA damage in CIII deficiency, we injected the *Bcs1l[p.S78G];mt-Cyb[p.D254N]* mice after weaning with recombinant adeno-associated viral vectors (rAAV) that deliver hepatocyte-specific expression of Omomyc (Fig. 10a), a dominant-negative mutant fragment of c-MYC[74]. Based on H2AX phosphorylation, Omomyc significantly decreased the DNA

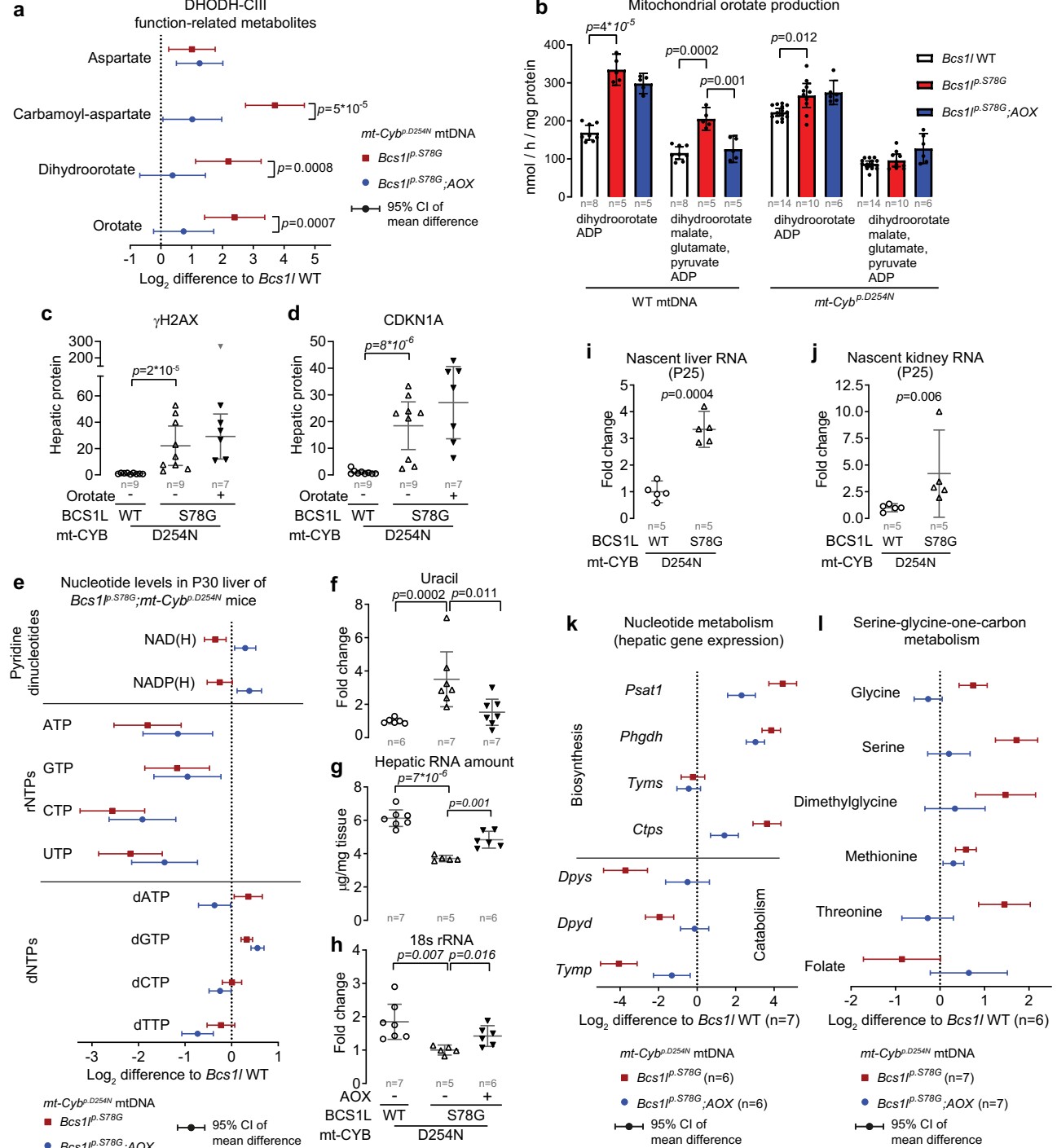

**Fig. 6 | CIII deficiency disturbs nucleotide pools and amplifies transcription and nucleotide biosynthesis. a** UPLC-MS quantification of key metabolites related to DHODH function in P30 liver tissue of *Bcs1l* WT (*n* = 6 for aspartate and *n* = 11 for other metabolites), *Bcs1l^p.S78G^* (*n* = 7 for aspartate and *n* = 11 for other metabolites), and *Bcs1l^p.S78G^*;AOX (*n* = 7 for aspartate and *n* = 11 for other metabolites). **b** Orotate production in isolated P30 liver mitochondria. For maximal CIII-linked orotate production, dihydroorotate and ADP served as respiratory substrates. For orotate production during CI-linked respiration, the assay contained malate, glutamate, and pyruvate in addition to dihydroorotate and ADP. **c, d** Effect of DHODH bypass via orotate supplementation (weaning to P29) on hepatic expression of γH2AX and CDKN1A in *Bcs1l^p.S78G^*;*mt-Cyb^p.S254N^* mice (Western blot). One outlier sample (gray upside-down triangle) was excluded from the calculation of mean and 95%-CI. **e** The levels of pyridine dinucleotides (*n* = 7 for *Bcs1l* WTs, *n* = 8 for *Bcs1l^p.S78G^*, *n* = 9 for *Bcs1l^p.S78G^*;AOX), rNTPs (*n* = 8 for *Bcs1l* WTs, *n* = 8 for *Bcs1l^p.S78G^*, *n* = 7 for

*Bcs1l^p.S78G^*;*AOX*), and dNTPs (*n* = 15 for *Bcs1l* WTs, *n* = 16 for *Bcs1l^p.S78G^*, *n* = 8 for *Bcs1l^p.S78G^*;*AOX*) in liver extracts. NAD(H), NADP(H), and dNTPs were measured using enzymatic assays and rNTPs using UPLC-MS. **f** Uracil concentration in P30 liver extracts. **g** Hepatic total RNA amount as a measure for ribosomal RNA depletion. **h** Hepatic 18S rRNA amount normalized to two reference mRNA transcripts (qPCR). **i, j** Quantification of ethynyl uridine-labeled nascent RNA from liver and kidney samples. **k** Hepatic expression of nucleotide biosynthesis- and catabolism-related genes. **l** Quantification of metabolites related to serine-glycine-one-carbon metabolism from liver extracts. All data are from one-month-old mice, except nascent RNA quantifications, for which P25 mice were used. Statistics: one-way ANOVA followed by the selected pairwise comparisons (Welch's *t*-statistics). The error bars in **b–d** and **f–j** represent 95% CI of mean. All data points derive from independent mice. Source data are provided as a Source Data file.

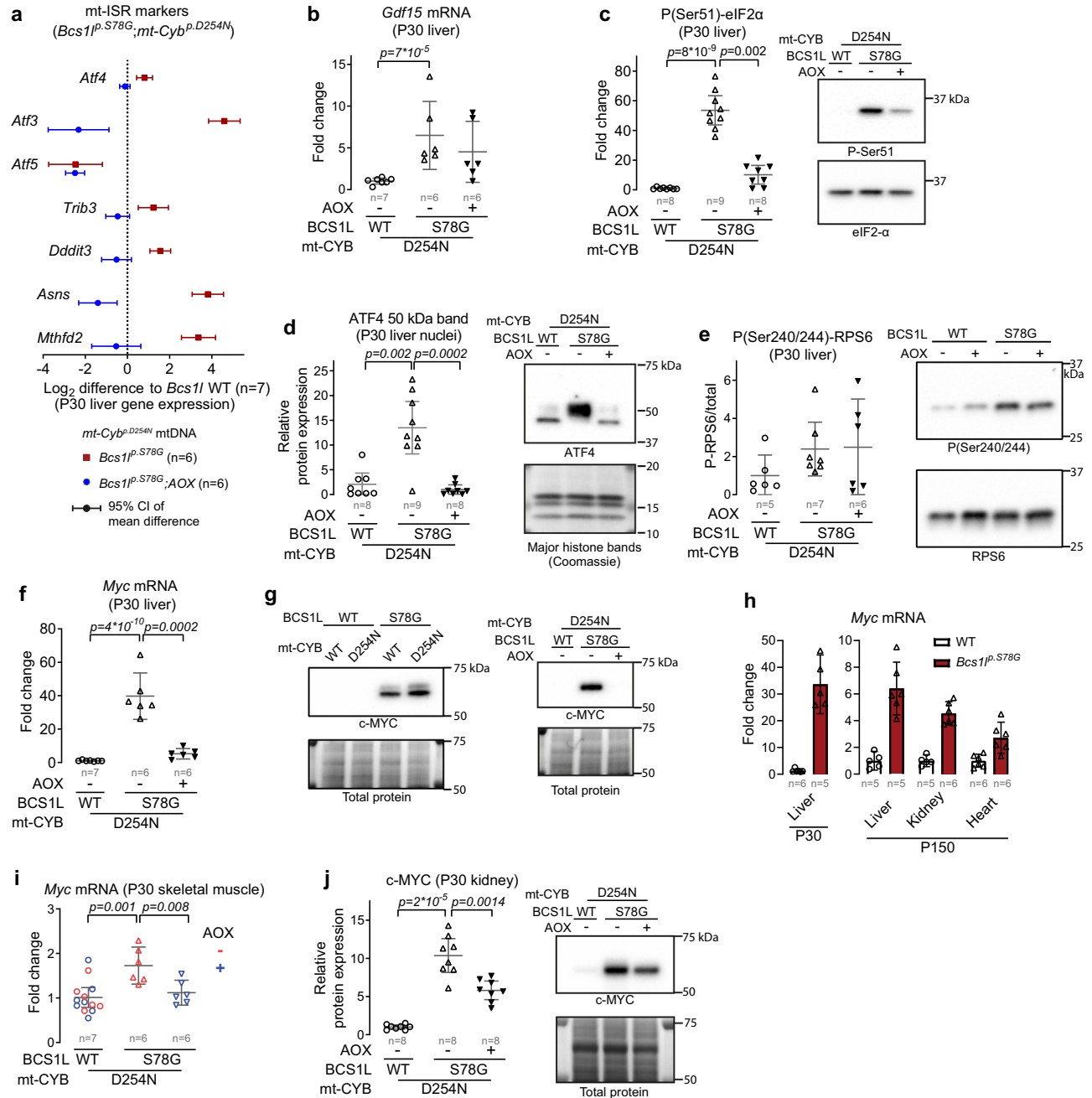

**Fig. 7 | AOX supresses mitochondrial integrated stress response (mt-ISR) and c-MYC induction. a** Gene expression of mt-ISR markers in P30 liver. **b**, *Gdf15* mRNA expression in P30 liver. **c, d** Western blot quantification of eIF2-α phosphorylation and ATF4 expression as readouts of mt-ISR activation. ATF4 was detected from liver nuclear fractions. **e** Western blot assessment of mTORC1 activation based on RPS6 phosphorylation status. **f** *Myc* mRNA expression (qPCR) in P30 liver from mice of *mt-Cyb*[p.D254N] background. **g** c-MYC protein expression in P30 liver from mice of the

indicated genotypes. **h** *Myc* mRNA in P30 liver (qPCR), and P150 liver, kidney and heart (RNAseq) from mice with WT mtDNA. **i** Myc mRNA expression in P30 skeletal muscle (quadriceps). **j** c-MYC protein expression in P30 kidney lysates. Statistics: one-way ANOVA followed by the selected pairwise comparisons (Welch's *t*-statistics). The error bars in **b–j** represent 95%-CI of mean. All data points derive from independent mice. Source data are provided as a Source Data file.

damage at level of whole liver (Fig. 10b, g). Although the main mechanism of action of Omomyc is to inhibit c-MYC binding to its target DNA sequences[75], Omomyc also decreased the endogenous c-MYC amount (Fig. 10c), as reported[76]. The endogenous c-MYC expression and γH2AX signal showed a very strong positive correlation (Fig. 10d, e). Given that DNA damage should downregulate c-MYC[77], this correlation suggests a linear dose response between c-MYC expression and the DNA damage. In non-transduced liver cells of Omomyc and control rAAV-injected mutant mice, the proportion of γH2AX-positive nuclei was roughly similar (Fig. 10f, h). In

Omomyc-transduced cells, the γH2AX staining was largely absent (Fig. 10f, i).

## Discussion
Overall, two unexpected conclusions arise from our data (Fig. 10j). First, non-replicating tissue parenchymal cells can be largely indifferent to severe OXPHOS deficiency, as shown by the AOX-expressing mice. Second, severe CIII deficiency, paradoxically, does not suppress energy-consuming anabolism nor cell cycle entry or progression to the Sphase, leading to catastrophic replication attempt. The *Bcs1l*[p.S78G] and

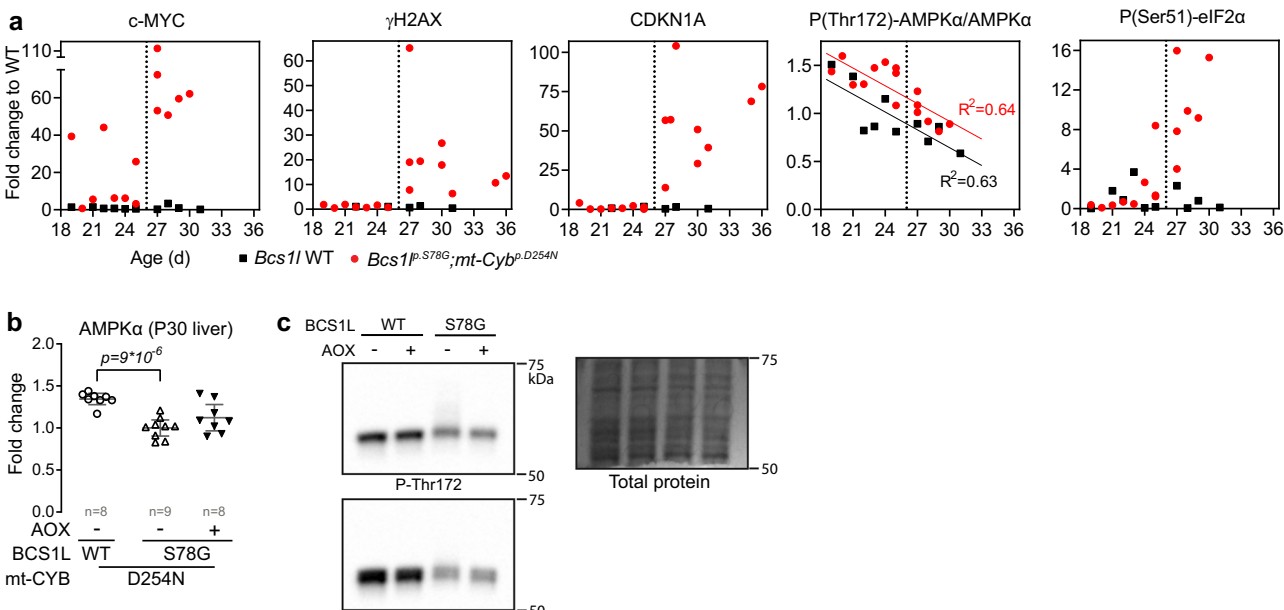

**Fig. 8 | c-MYC upregulation precedes the DNA damage and cell cycle arrest.**
**a** Hepatic c-MYC, CDKN1A, and γH2AX expression, and the phosphorylation status of AMPKα and eIF2α plotted against mouse age (Western blot quantifications). **b**, **c** Total amount AMPKα in P30 liver lysates and a representative Western blot.

Statistics: one-way ANOVA followed by the selected pairwise comparisons (Welch's *t*-statistics). The error bars represent 95%-CI of mean. All data points derive from independent mice. Source data are provided as a Source Data file.

*Bcs1l^{p.S78G};mt-Cyb^{p.D254N}* mice, which closely recapitulate the neonatal GRACILE syndrome, demonstrate that the ultimate outcome of persistent proliferation against CIII deficiency is a juvenile-onset segmental progeria. The first signs of premature aging in *Bcs1l^{p.S78G}* mice appear before 1 month of age. This onset is similar to that in homozygous *Lmna^{p.G609G}* knock-in mice[78] and 2 weeks earlier than in *Zmpste24* knock-out mice[79,80], two established mouse models of HGPS. *Bcs1l^{p.S78G}* mice and these two mouse models of HGSP have roughly similar lifespan (4-7 months)[25,26], whereas the rapid deterioration of *Bcs1l^{p.S78G};mt-Cyb^{p.D254N}* mice, with a lifespan less than 40 days, makes them one of the most severe mouse models of accelerated aging. Strikingly, the liver histology and transcriptome changes of *Bcs1l^{p.S78G}* mice were almost a genocopy of the ERCC1-deficient mice, one of the most widely characterized progeroid mouse models[40,49,81].

Thus far, premature aging has largely been a peculiarity of the artificial *Polg^{p.D257A}* mtDNA mutator mice among mouse models of OXPHOS deficiency[7]. However, phenotypes that overlap with segmental progerias, such as short stature, progressing emaciation, hearing loss, anemia, alopecia, and skin lesions are relatively common among OXPHOS disorders including some CIII deficiencies[82–85]. Moreover, leukocytes from a patient with CIV deficiency show Fanconi anemia-like response to DNA damage[86]. Later studies found increased nuclear DNA damage and evidence of impaired DNA repair in skin fibroblasts of the same patient and in fibroblasts with induced CIV deficiency[87,88]. Intriguingly, mutations in *SLC25A24*, encoding one of the mitochondrial ATP-Mg²⁺-Pi-transporters, can cause Fontaine syndrome with similarity to HGPS[89,90]. Also, mutations in the mitochondrial outer membrane proteins MTX2 and TOMM7 were recently linked to progeroid syndromes[91,92]. There is very little knowledge of the underlying mechanisms. A recent study proposed a novel[7], yet debated[93,94], mechanism whereby increased mtDNA repair distinctively in the mutator mice consumes dNTPs excessively, leading to replication stress in some highly proliferative cell types. Our data from the *Bcs1l^{p.S78G}* mice, however, show that isolated CIII deficiency is sufficient to cause nuclear DNA damage and segmental progeria.

Altered mitochondrial function is a feature of cellular senescence of various etiologies[32]. On the other hand, a number of studies have shown that mitochondrial dysfunction is sufficient to cause cellular senescence or related conditions such as replication stress and cell cycle arrest in cultured cells[7,11,16,52,56,57]. Furthermore, BCS1L and another CIII assembly factor LYRM7 have emerged as positive hits in genome-wide RNA interference screens with nuclear DNA damage as a readout[95–97]. The previous studies have each suggested a single mechanism, such as compromised NAD(H) redox status or pyrimidine biosynthesis, as an explanation. Our study is the first one to comprehensively examine the metabolic alterations underlying senescence due to a primary OXPHOS deficiency in vivo. Our data depict a scenario in which metabolic checkpoint evasion and proliferation against depletion and imbalance of multiple mitochondria-dependent metabolites, particularly strikingly rNTPs, lead to replication stress, DNA damage, and cellular senescence. Intriguingly, we found this to mainly affect differentiated cells with replicative potential such as hepatocytes and renal tubular epithelial cells, explaining the unusual visceral pathology in this model. Counterintuitively, the highly proliferating compartment bone marrow avoided the DNA damage, probably simply because the loss of CIII activity due to the *Bcs1l^{p.S78G}* mutation is not severe enough in this tissue. Notably, however, complete loss of CIII activity in mouse hematopoietic stem cells via conditional RISP knockout causes aberrant cell cycle entry and induction of γH2AX in these and the derived progenitor cells[98]. On the other hand, while the loss of CIII activity in skeletal muscle due to the *Bcs1l^{p.S78G}* mutation is as severe as in the liver[25], the myocytes avoided the DNA damage because of the terminal post-mitotic state of these cells, according to our interpretation. Supporting our conclusions, and in line with our in vitro data from AML12 cells, proliferation was reported to be a prerequisite to increased DNA damage in CIV-deficient human skin fibroblasts[87].

Astonishingly, bypass of the CIII-CIV segment with transgenic AOX expression completely prevented the hepatic nuclear DNA damage response despite that it left the canonical detrimental consequences of OXPHOS dysfunction largely untouched. Instead of correcting the primary metabolic defects, AOX limited the cell cycle

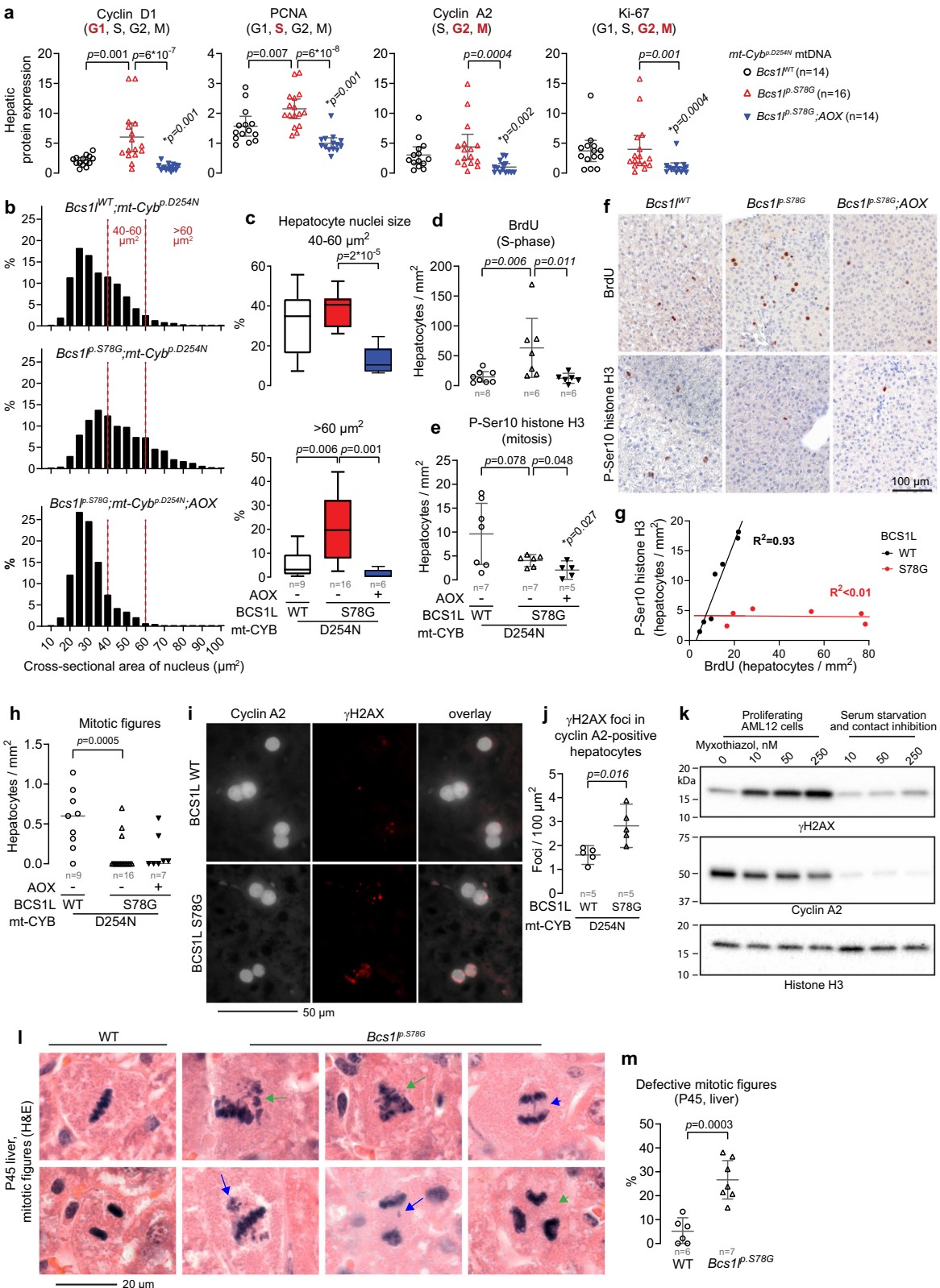

entry of normally quiescent differentiated cells. Most likely this considerably delayed the cellular senescence and cell death due to replication stress, allowing extended survival of the *Bcs1l^p.S78G^;mt-Cyb^p.D254N^* mice. Scarcity of biosynthetic precursors typically limits cellular proliferation, but, unexpectedly, this was not the case in the liver and kidney of *Bcs1l^p.S78G^;mt-Cyb^p.D254N^* mice, which attempted to grow or regenerate despite the lack of resources. Similar illicit cell

cycle entry and cell cycle progression is a common feature of cancer cells, a large percentage of which overexpress c-MYC[66]. In cancer, but also in normal cells, excessive c-MYC potently promotes proliferation, facilitates evasion of metabolic checkpoints, accelerates the S-phase entry and duration, and causes replication stress especially when the resources do not meet the requirements[66,70]. Considering the disproportionately increased c-MYC expression, which was at

**Fig. 9 | CIII deficiency leads to illicit cell cycle entry, replication stress, and mitotic catastrophe. a** Hepatic expression of the indicated proliferation markers (Western blot analyses). The cell cycle phases in which each marker protein is expressed is marked in parentheses with bold font indicating the phase(s) with the peak expression. The samples were run in a randomized order on two 26-well gels. *p*, comparison to *Bcs1l* WT. Supplementary Fig. 8a shows the representative blots. **b** Hepatocyte nuclei size profile. The histograms comprise 3352 nuclei from 9 *Bcs1l* WTs, 4131 from 16 *Bcs1l^{p.S78G}* mice, and 2980 from 6 *Bcs1l^{p.S78G};AOX* mice. **c** The proportion of large (40–60 $\mu m^2$) and atypically large hepatocyte nuclei (>60 $\mu m^2$). **d** Number of bromodeoxyuridine (BrdU) -positive hepatocytes as a measure of cells in the S-phase. **e** Number of mitotic cells based on histone H3 phosphorylation (P-Ser10). **f** Representative immunostainings for BrdU and P-Ser10 H3. **g** Correlation between BrdU- and histone H3 P-Ser10 -positive hepatocytes (S-phase vs mitosis). **h** Quantification of mitotic figures from H&E-stained liver sections. **i, j** γH2AX foci in cyclin A2-positive hepatocytes as a measure of DNA damage in S-, G2-, and early mitosis of the cell cycle. **k** Effect of the CIII inhibitor myxothiazol on γH2AX levels in AML12 cells in growth phase and under suppression of proliferation by serum starvation and contact inhibition (representative result of more than 3 similar experiments). **l** Representative defective mitotic figures in *Bcs1l^{p.S78G}* hepatocytes. Green arrows, multipolar mitotic spindles; green arrowhead, multipolar anaphase; blue arrows, lagging or aberrantly dispersed chromosomes or chromatin fragments; blue arrowhead, anaphase bridge. **m** Quantification of defective mitotic figures. All mouse data are from P30 mice unless otherwise shown in the figure. Statistics: **a, c, d, e** one-way ANOVA followed by the selected pairwise comparisons (Welch's *t*-statistics); **h** Kruskal–Wallis test followed by Mann–Whitney *U* test. **j, m** Welch's *t*-test. The error bars in scatter blots represent 95% CI of mean. The box blots show the median, the quartiles, and the minimum and maximum. All data points derive from independent mice. Source data are provided as a Source Data file.

least an order of magnitude higher than in liver injury models such as 2/3 hepatectomy and bile duct ligation[99,100], a similar scenario is inevitable to take place in the replicating cells of the affected tissues in the *Bcs1l^{p.S78G}* mutant mice. However, unlike cancer cells that have evolved to tolerate replication stress, the hepatocytes apparently succumb to mitotic catastrophe or cellular senescence. Indeed, suppression of c-MYC function with the dominant-negative Omomyc protein proved sufficient to alleviate the DNA damage in the CIII-deficient liver. Intriguingly, the c-MYC homolog in yeast is Retrograde regulation protein 1 (Rtg1), which conveys a mitochondrial stress retrograde signal to the nucleus[101]. Some evidence suggests a partly analogous retrograde signal in mammalian cells[102]. In line with this hypothesis, and in addition to our data, transcriptome studies have shown *Myc* upregulation in OXPHOS-defective mouse models[51,62,65] and patient cell lines[64]. Furthermore, in the fruit fly, mt-ISR can promote aberrant hyperproliferation[103]. However, *dMyc* expression data was not reported. Our study provides evidence for a role for c-MYC upregulation in the pathogenesis of OXPHOS deficiency and warrants further studies on this largely neglected putative mitochondrial retrograde signal in mammalian cells. Genetic experiments to manipulate MYC, for example crossing of the *Bcs1l^{p.S78G}* mice with *c-Myc* knock-out or a hypomorphic allele-carrying mice, would undoubtedly be very interesting. Furthermore, the universality and the underlying triggers of c-MYC induction upon OXPHOS deficiency remain yet to be clarified. One interesting possibility is that the c-MYC induction is downstream of the EGFR ligand AREG, a SASP-related growth factor and a recently identified mitokine[104].

AOX robustly altered mitochondrial stress signaling, affecting metabolic checkpoints or growth factor signaling, and importantly, preventing the c-MYC upregulation and the deleterious cell cycle entry against CIII deficiency. This could be related to as yet unidentified coenzyme Q pool redox surveillance mechanism, or to the decreased mitochondrial membrane potential, or to the further suppression of the already decreased mitochondrial ROS production. Cellular $H_2O_2$ concentration is known to either promote or suppress proliferation of various cultured cells depending on its concentration[105]. Moreover, mitochondrial ROS positively regulates AMPK activation[106], which likely explains the apparently insufficient AMPK activation in *Bcs1l^{p.S78G}* liver. Astonishingly, a very different intervention compared to AOX, ketogenic diet, markedly decreased c-MYC expression and γH2AX levels in the *Bcs1l^{p.S78G}* mice. Intriguingly, ketogenic diet has been reported to restore hair growth, for the duration of the diet, in a child with CIII deficiency[107].

Together with the literature, our findings suggests that progeroid manifestations may be an unrecognized feature of other OXPHOS disorders. The principal effect on replicating cells is a key converging mechanism with several known progeroid syndromes[44]. However, the OXPHOS-linked etiology entails that the presence or absence of progeroid manifestations in a given mitochondrial disease is dictated by the highly cell- and cell cycle stage-dependent requirements for OXPHOS and sensitivity to its dysfunction. Such diseases would not necessarily affect proliferating cell types that are not metabolically dependent enough on OXPHOS or do not manifest the OXPHOS defect. The AOX and ketogenic diet interventions are a proof of concept that mitochondrial progerias can be treatable. Our findings urge further studies on the role of c-MYC, aberrant cell proliferation, and accumulation of senescent cells in mitochondrial diseases.

## Methods

### Mouse lines and husbandry
We have described the *Bcs1l^{p.S78G}*, *mt-Cyb^{p.D254N}*, and AOX mouse lines in detail in our previous publications[23,25,26]. *Bcs1l* WTs and heterozygotes, irrespective of the *mt-Cyb^{p.D254N}* variant, were considered phenotypically WT. The analyses included both sexes unless otherwise stated. Where appropriate, the figures show the data separately for both sexes. All mice were on C57BL/6JCrl (RRID:IMSR_JAX:000664) nuclear genomic background and received water and chow (Teklad 2018, Harlan) *ad libitum*. The animal facilities of University of Helsinki housed the mice in temperature-controlled (23 °C) individually-ventilated cages under 12-h light/dark cycle. The animal ethics committee of the State Provincial Office of Southern Finland approved the animal studies (permit numbers ESAVI/6365/04.10.07/2017 and ESAVI/16278/2020). We performed the animal experiments according to the FELASA (Federation of Laboratory Animal Science Associations) guidelines and best practices. The AOX transgene is intellectual property of Tampere University, Finland. All other mouse lines are available from the corresponding author upon request.

### In vivo labeling of nascent DNA and RNA
Replicating cells and nascent mtDNA were labeled by administering a dose of 100 mg/kg body weight BrdU (Sigma-Aldrich, #B5002) via intraperitoneal injection to P28-29 mice. Tissue samples were collected 16 h later. The samples from BrdU-exposed mice were not used for DNA damage assays.

Nascent RNA was labeled by intraperitoneal injection of 88 mg/kg 5-ethynyl-uridine (EU) (Jena Biosciences, #CLK-N002-10) to P25 mice. Tissue samples were collected 6 h later.

### Dietary interventions
Orotate-supplemented chow (1% w/w) was prepared by adding orotic acid (Sigma-Aldrich, #O8402) to moistened chow followed by pressing the chow into pellets. These pellets were dried at room temperature and stored at −20 °C until use. The treated mice received this chow *ad libitum* from weaning (P19-22) until sample collection at P29.

The ketogenic diet intervention has been described in detail previously[27]. Briefly, we utilized here samples from mice fed control

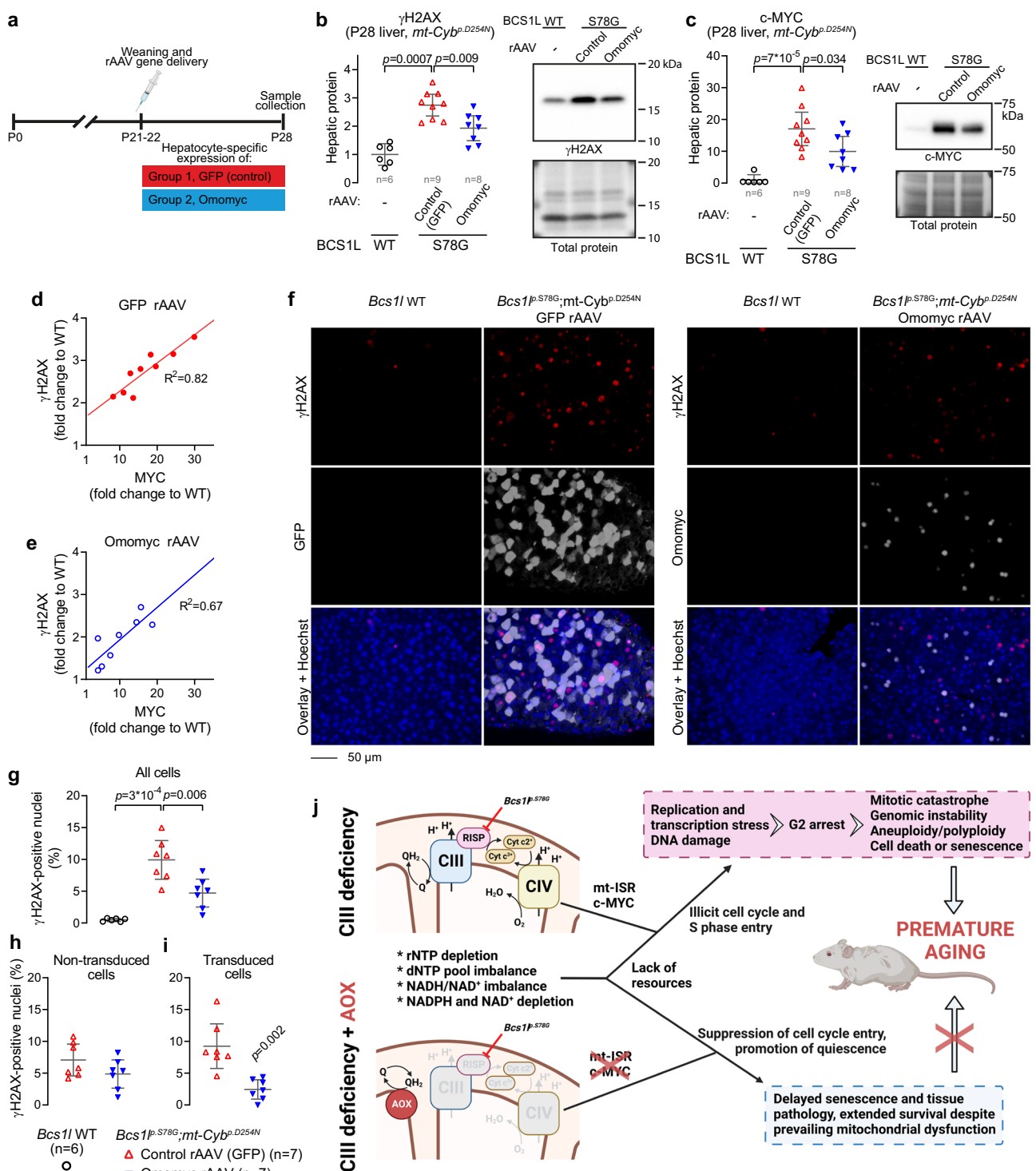

**Fig. 10 | Suppression of c-MYC function with dominant-negative Omomyc protein attenuates the CIII deficiency-induced DNA damage. a**, Timeline of rAAV-mediated expression of Omomyc in the hepatocytes of *Bcs1l^{p.S78G}*;*mt-Cyb^{p.D254N}* mice. The control rAAV expressed the green fluorescent protein (GFP) as a reporter protein. **b**, **c** Western blot quantification of γH2AX and c-MYC in liver lysates from *Bcs1l WT* and rAAV-injected *Bcs1l^{p.S78G}*;*mt-Cyb^{p.D254N}* mice. **d**, **e** Correlation between c-MYC expression and H2AX phosphorylation in the liver samples from *Bcs1l^{p.S78G}*;*mt-Cyb^{p.D254N}* mice injected with control rAAV (**d**) or Omomyc rAAV (**e**). The data are same as in the figures b and c. **f** Representative liver sections

immunostained for γH2AX and either GFP or Omomyc. **g–i** Quantification of γH2AX-positive nuclei in liver sections: **e** both transduced and non-transduced cells, **f** non-transduced cells, **g** transduced cells. **j** Schematic presentation of the proposed mechanisms leading to premature aging of *Bcs1l^{p.S78G}* mice and the effect of AOX. Statistics: **b**, **c**, **g** one-way ANOVA followed by the selected pairwise comparisons (Welch's *t*-statistics); **i** Welch's two-sided *t*-test. The error bars represent 95% CI of mean. All data points derive from independent mice. Source data are provided as a Source Data file.

chow or ketogenic diet (TD.96355, Harlan) from weaning until approximately P45.

## rAAV cloning, production, and administration

For the cloning of the Omomyc rAAV construct, a human c-MYC fragment encoding amino acids 350–439 was PCR-amplified (5′ primer introduced a Kozak sequence and start codon ATG), cloned into a pBluescript vector, and the four Omomyc mutations (E410T, E417I, R423Q, R424N) introduced by inverse PCR mutagenesis. After sequencing, the fragment was subcloned into the pAAV2.LSP1 vector (a kind gift from prof. Ian Alexander, University of Sydney)[108]. This vector delivers hepatocyte-specific expression under human ApoE enhancer and α1-antitrypsin promoter. A vector encoding enhanced green fluorescent protein was used as control. Serotype 9 viral particles were produced by AAV Gene Transfer and Cell Therapy Core Facility of University of Helsinki. The treated mice received $5 \times 10^{10}$ viral particles in $100 \, \mu l$ saline via intraperitoneal injection at P21–22. The tissue samples were collected on P28. The Omomyc and GFP expression were verified by Western blotting and immunohistochemistry on paraffin sections. The rabbit polyclonal Omomyc antibody was a kind gift from prof. Laura Soucek's laboratory (Vall d'Hebron Institute of Oncology, Barcelona, Spain).

## Sample collection

For all metabolite analyses, the mice were euthanized by cervical dislocation and the left lateral lobe of the liver collected into liquid nitrogen within a few seconds. For other analyses, the mice were euthanized similarly or by pentobarbital and cardiac puncture and the tissue samples snap frozen in liquid nitrogen, or immersion fixed for histology, or used for analyses requiring fresh tissue. Before sample collection, the mice were fasted for 2 h. All comparable samples were collected at the same time of the day during the light period of the mice.

## Body composition analyses

Fat mass was quantified using echoMRI-based MiniSpec Body Composition Analyzer from Bruker. Hindlimb bone mineral density was quantified using Dual-Energy X-ray Absorptiometry (DEXA) (Lunar PIXImus 2 Mouse Densitometer equipment, GE Medical Systems). The same equipment was also used to obtain images for the quantification of thoracic-lumbar curvature. Thymi were weighted after fixation in 10% histological formalin (~48 h). Spleens were fixed similarly and weighted after transfer into 70% EtOH.

## Cell culture

AML12 cells (ATCC, # CRL-2254) were cultured in DMEM:Ham's F-12 medium with 10% fetal bovine serum, 5 mM glucose, glutamine, an insulin-transferrin-selenium supplement (Gibco), and penicillin and streptomycin. The chemical treatments (myxothiazol, antimycin A) were performed in 12-well plates for 20–24 h with or without serum starvation. For the serum starvation experiments, the cells were grown to near confluence, and then grown without the serum and insulin-transferrin-selenium supplement for at least 16 h, after which the medium was replaced with fresh serum-free medium and the chemicals added for 20–24 h.

## SDS-PAGE, blue-native PAGE, and Western blot

Protein concentrations were measured with Bradford reagent (Bio-Rad) and bovine serum albumin standards after an appropriate sample dilution and, if necessary, inclusion of 2.5 mg/ml α-cyclodextrin in the reagent to chelate interfering sodium dodecyl sulfate (SDS).

For the analysis of albuminuria and MUPs, urine samples were boiled in Laemmli sample buffer (LSB) and volume equivalent to $1 \, \mu l$ urine separated in 4–20% tris-glycine PAGE. The gels were stained with Coomassie G-250 and imaged with a flatbed scanner.

For denaturing Western blot analyses, tissue samples or cultured cells were homogenized in bromophenol blue-free LSB supplemented with 1 mM EDTA and protease inhibitors. Sample viscosity was decreased by a brief sonication or passing the sample through a 27 g syringe. SDS-PAGE was performed on Bio-Rad's mini- or midi-sized gels with an appropriate protein loading (5–30 μg) and polyacrylamide percentage for the target proteins. Transfer onto PVDF or nitrocellulose membranes were performed with tank transfer method and standard Towbin transfer buffer containing 0–20% MeOH as optimal for the target proteins. Equal loading and transfer were verified by staining the membranes with Coomassie G-250 (PVDF) or Ponceau S (nitrocellulose). Supplementary Table 1 lists all antibodies used. Peroxidase-conjugated secondary antibodies, enhanced chemiluminescence, and Bio-Rad ChemiDoc MP Imaging System and Image Lab software were used to detect and document the immunodetections. All samples subjected to quantification were processed and run in a randomized order. For representative blots, individually analyzed samples were pooled together from each experimental group to obtain a representative average signal.

Digitonin solubilization of respiratory complexes and blue-native PAGE analyses were performed as described[25].

## Redox Western blot analyses

Approximately 10 mg frozen liver samples were homogenized in $400 \, \mu l$ of buffer comprising 60 mM Tris-Cl pH 6.8, 12.5% glycerol, 1 mM EDTA, protease inhibitors (Roche, cOmplete™ protease inhibitor mix), 10 μg/ml catalase, and 50 mM N-ethylmaleimide to block spontaneous thiol oxidation. After equalization of proteins concentrations and addition of 2% SDS to denature proteins, the lysates were sonicated and subjected to non-reducing Western blot. To minimize bias in epitope availability between oxidized and reduced peroxiredoxins, the blotted proteins were subjected to on-membrane disulfide cleavage (1% β-mercaptoethanol in Tris-buffered saline) followed by thiol alkylation with 5 mM N-ethylmaleimide (in Tris-Cl buffer pH 6.8).

## RNAseq and qPCR

The P150 liver and kidney RNA-seq transcriptomics have been described previously and the data are available from the ArrayExpress database under the accession ID E-MTAB-7416[26]. Heatmaps were generated using Gitools 2.3.1 (http://www.gitools.org).

For qPCR analyses, we prepared cDNA and employed EvaGreen- and Phire II Hot Start DNA polymerase-based detection chemistry as described[25]. The qPCR reactions were carried out in Bio-Rad CFX96 or CFX384 instruments and the data analyzed with either Bio-Rad CFX manager or CFX maestro software. Genomic DNA was isolated using a typical SDS-proteinase K digestion followed by phenol-chloroform purification and EtOH precipitation. Mitochondrial mtDNA content was quantified by employing direct qPCR on mitochondrial suspensions boiled in 1% SDS and then diluted 1:50,000 with water. *Gak* and *Rab11a* served as reference genes for the normalization of mRNA expression data, unless otherwise stated. All qPCR data were corrected for PCR efficiency. For quantification of *progerin*, we tested 3 reverse primers spanning the *progerin*-specific 150 nt deletion. The optimal primer was predicted to align to *lamin A* transcript with a 1-nt gap at the third nucleotide from the 3′-end or with three consecutive 3′-mismatches. A gradient qPCR showed an optimal annealing temperature of 66 °C with our qPCR reagents and 500 nM concentration of each primer. For some runs, we employed a plate reading step at elevated temperature (+81 °C) to eliminate unspecific late-cycle primer-dimer-like signal in the absence of specific template. The *progerin* qPCR was validated by running the products on agarose gel and by their Sanger sequencing. Primer sequences are provided in Supplementary Table 2.

## Histology

Tissue sections from formalin-fixed paraffin-embedded samples were stained with standard methods to assess general histology (H&E), and morphology of nuclei (Hoechst and H&E). Lipofuscin autofluorescence was imaged from deparaffinized and rehydrated sections mounted with Tris-buffered 90% glycerol (pH 8.3)[27].

For immunohistochemical staining of paraffin sections, Vectastain Elite ABC peroxidase or alkaline phosphatase reagents, or ImmPRESS peroxidase or alkaline phosphatase polymer detection reagents (Vector Laboratories) were used. Fluorescent tyramide CF dye conjugates (Biotium) were used as peroxidase substrates in the double stainings. Antibodies are listed in Supplementary Table 1.

β-galactosidase activity staining was performed on 14 μm cryosections fixed for 10 min with 0.2% glutaraldehyde in PBS. The staining solution comprised 0.5 mg/ml X-Gal, 5 mM potassium ferrocyanide, 5 mM potassium ferricyanide, 130 mM NaCl, 2 MgCl$_2$, and 43 mM citrate buffer pH 4.25. The staining was stopped after 2 h incubation at 37 °C. The sections were post-stain fixed with 10% formalin and faintly counterstained with Mayer's hematoxylin.

Acquisition of images was performed with Zeiss Axio Imager M2 microscope, Axiocam 503 camera, and Zen 2.3 software.

Preparation of the electron microscopy samples have been described in our previous publications[26,27].

## Cell counting and image quantifications

Apoptotic and necrotic cells (single-cell necrosis) were counted from H&E-stained sections based on characteristic morphologic features such as condensed hypereosinophilic cytoplasm, cell boundary halo, cell swelling, and condensed, fragmented, or dissolved nucleus[27]. Misshapen hepatocyte nuclei were counted from H&E-stained sections and electron micrographs based on the presence of nuclear envelope blebs, invaginations, overall wavy nuclear envelope appearance, or atypical nuclear shape. The count included nuclei with inclusion body-like structures. Cells showing morphological features of apoptosis or necrosis were excluded. For quantification of ductular reactions, we included all ductules outside portal triads and those portal ductules that exceeded the typical 2 duct-like structures visible per portal triad cross section. The size of hepatocyte nuclei was estimated by measuring the diameter of nuclei and assuming the shape of circle.

Automated image quantification was applied for lipofuscin, BrdU, γH2AX, Omomyc, and GFP. Lipofuscin autofluorescence was quantified using ImageJ Fiji software (National Institutes of Health, Bethesda, MD, USA) thresholding tool (same threshold for all images) to select the positive area. Integrated fluorescence density of the positive area of image was then taken as a measure of relative lipofuscin amount per the imaged cross section. For automated quantification of colorimetric stains, shading correction based on several averaged blank pictures was applied. Color deconvolution was performed using Photoshop CC (Adobe Inc.) Color Range tool. ImageJ Fiji Threshold and Analyze Particles tools were employed to count the number of positive cells or nuclei, or confine further analyses to the relevant nuclei. For quantification of BrdU-positive hepatocyte nuclei and γH2AX foci, the nuclei of other cell types were filtered out based on nuclei size and shape (circularity). In quantification of γH2AX in rAAV-transduced and non-transduced cells, Hoechst signal was utilized to count the number of nuclei using ImageJ Fiji Find Maxima function after manual exclusion of large clusters of small non-parenchymal cells and applying Gaussian blur filter to smoothen the nuclear stain. For all quantifications, most of the tissue section was covered by taking 5–50 non-overlapping pictures with an appropriate magnification. The quantifications were performed blindly as for the sample genotype.

## Mitochondrial and nuclear fractions

Liver and kidney mitochondria were isolated using previously described differential centrifugation protocol[25]. To obtain liver nuclear fraction, 40 mg tissue was homogenized in 1.4 ml of isotonic buffer (10 mM HEPES-NaOH pH 7.4, 75 μg/ml digitonin, 250 mM sucrose, and 5 mM Mg acetate) containing protease, phosphatase, and deacetylase inhibitors (Roche c0mplete protease inhibitor mix, 10 mM NaF, 1 mM Na$_3$VO$_4$, 1 mM EGTA, 1 μM trichostatin A, and 1 mM Na-butyrate). The homogenate was pass through a 70-μm cell strainer and centrifuged 600 × g for 10 min at +4 °C. The pelleted nuclei were resuspended in buffer comprising 0.2% Triton X-100, 40 mM NaCl, 5 mM Mg acetate, 10 mM HEPES-NaOH pH 7.4, and the protease, phosphatase, and deacetylase inhibitors. After re-centrifugation, the nuclear proteins were solubilized by sonication and boiling in LSB.

## Quantification of BrdU-labeled mtDNA

We employed in vivo BrdU-labeling and two different DNA extraction methods to assess the rate of mtDNA synthesis. In the first method, frozen liver samples were homogenized with glass-teflon Potter-Elvehjem in buffer comprising 225 mM mannitol, 75 mM sucrose, 10 mM Tris-HCl pH 7.4, 1 mM EGTA, 2.5 mM EDTA and 50 μg/ml RNAse A. The nuclei were pelleted by a 800 × g centrifugation for 10 min at +4 °C and the supernatant collected. This centrifugation step was repeated once more followed by a 14,000 × g centrifugation to pellet the mitochondria. DNA from the mitochondrial fractions were isolated using SDS-proteinase K digestion, phenol-chloroform purification, and EtOH precipitation. In the second method, we isolated mtDNA from postnuclear fractions with the alkaline denaturation-neutralization procedure (Nucleospin Plasmid Mini kit, Macherey-Nagel). Liver tissue was homogenized and nuclei pelleted (1000 × g 10 min) in buffer comprising 25 mM HEPES-NaOH pH 7.4, 1 mM EDTA, 1 mM deferoxamine, 0.1% Triton X-100, and 50 μg/ml RNAse A. The extraction was continued with the alkaline lysis according to the manufacturer's protocol. DNA concentrations were measured with AccuBlue® Broad Range dsDNA Quantitation Kit (Biotium).

DNA from mitochondrial fractions were dot blotted using Bio-Rad microfiltration apparatus onto nylon membrane in the presence of 0.4 M NaOH to denature the DNA. The membranes were baked for 30 min at 80 °C followed by blocking with 5% milk in TBST. BrdU was detected using a mouse monoclonal antibody Bu20a (Dako), peroxidase-conjugated secondary antibody, and chemiluminescence. Results from the two mtDNA extractions are presented as geometric mean of signal intensities.

## Quantification of EU-labeled nascent RNA

Liver and renal cortex RNA from EU-exposed mice were extracted using RNAzol RT reagent (Sigma-Aldrich, # R4533). The incorporated EU was detected by click chemistry biotinylation and dot blotting. The click chemistry reactions comprised RNA and excess of 25 μM picolyl-azide-PEG4-biotin, 50 μM copper (II) sulfate, 200 μM BTTAA, and 5 mM ascorbate in PBS. This solution was made anoxic by bubbling with N$_2$ gas before use. The reactions were allowed to proceed for 40 min in dark with airspace filled with pure N$_2$. Equivalents of 50 ng reacted RNA were dot blotted in triplicates on Hybond N + nylon membrane (Amersham) using Bio-Rad Microfiltration apparatus. The membrane was dried for 30 min at 80 °C, rehydrated, and blocked with 0.5% polyvinyl alcohol and 0.5% polyvinylpyrrolidone in washing buffer (Tris-buffered saline, 1 mM EDTA, 0.2% tween-20). Streptavidin peroxidase and chemiluminescence were employed to detect the biotinylated EU-RNA.

## Mitochondrial respirometry

Oxygen consumption by isolated mitochondria was measured with Oxygraph-2k respirometer and DatLab 6 software (OROBOROS Instruments). The measurements were conducted in Mir05 buffer (110 mM sucrose, 60 mM lactobionic acid, 20 mM taurine, 20 mM HEPES, 10 mM KH$_2$PO$_4$, 3 mM MgCl$_2$, 0.5 mM EGTA, and 1 g/l fatty acid-free BSA, pH 7.1) at 37 °C. CI-linked phosphorylating respiration took

place in the presence of 1 mM malate, 5 mM pyruvate, 5 mM glutamate, 10 μM cytochrome *c*, and 1.25 mM ADP-Mg$^{2+}$. Following this, 10 mM succinate was added to measure CI&CII-linked phosphorylating respiration. AOX-mediated respiration was determined by the addition of 50 μM propyl gallate.

## Mitochondrial ATP production assay

Isolated liver mitochondria (12.5 μg protein) were incubated in 0.2 ml Mir05 buffer as described for CI-linked respiration but with inclusion of adenylate kinase inhibitor (25 μM P1,P5-Di(adenosine-5′)penta-phosphate). After a 20-min incubation at +37 °C, enzymatic activity was quenched by the addition of an equal volume of buffered phenol (pH 8) and by freezing (−80 °C) the samples. On the day of ATP measurement, phase separation was induced by addition of 100 μl chloroform and centrifugation (6 min, 15,000 × *g*). The aqueous phase was diluted 1:1000 with 25 mM Tris-acetate buffer pH 7.75 and ATP measured using luciferin-luciferase reagent (Invitrogen, #A22066). Background was estimated by omission of the incubation step. The luminescence was measured with Synergy H1 plate reader and Gen 3.08 software (BioTek). The same equipment and software were utilized for other microplate assays as well.

## Mitochondrial membrane potential

Tetramethylrhodamine methyl ester (TMRM) (Invitrogen) in self-quenching mode (1.5 μM) was utilized to assess the membrane potential of isolated liver mitochondria. The assay was performed at 37 °C in 96-well format. Mir05 buffer with 1 mM malate, 5 mM pyruvate, 5 mM glutamate, and 1.25 mM ADP served as the assay media. Maximal and fully collapsed membrane potential, as induced by oligomycin A (0.5 μg/ml) and SF6848 (0.5 μM), respectively, served as reference states.

## Mitochondrial H$_2$O$_2$ production

We utilized Amplex UltraRed peroxidase assay to measure mitochondrial H$_2$O$_2$ production. To increase the sensitivity of the assay and eliminate interference due to endogenous antioxidant systems and interfering carboxylesterases[55], we took the following measures. The mitochondria were exposed to 35 μM CDNB (1-chloro-2,4-dinitrobenzene) and 50 μM PMSF (phenylmethanesulfonyl fluoride) to deplete glutathione and irreversibly block carboxylesterases, respectively, during the isolation of mitochondria (exposure time 12 min on ice). These chemicals were washed away during the isolation process. Catalase and thioredoxin system were inhibited by inclusion of 100 μM 3-amino-1,2,4-triazole and 0.5 μM auranofin in the assay buffer (Mir05), respectively. Amplex UltraRed (Invitrogen) and horseradish peroxidase concentrations were 15 μM and 1 U/ml, respectively. Superoxide dismutase (5 U/ml) was included to ensure the immediate dismutation of superoxide to H$_2$O$_2$. Background was measured in the absence of exogenous substrates required for the mitochondrial respiratory activity. CI-linked respiration was stimulated by the addition of 1 mM malate and 5 mM glutamate. For CII- and CI&CII-linked respiration, succinate concentration was 10 mM. State 3 respiration was induced by the addition of 1.6 mM ADP. For every condition, parallel reactions were spiked with a 50 pmol bolus of H$_2$O$_2$ ($\varepsilon_{240\ nm}$ = 43.6 M$^{-1}$ cm$^{-1}$) to calibrate the fluorescence signal. The assay was performed in 96-well format (0.3 ml reactions) with short incubation times (5 min) to minimize fluctuations in O$_2$ concentration.

## Mitochondrial orotate production

Liver mitochondria (30 μg) were incubated in 150 μl Mir05 buffer containing 0.5 mM dihydroorotate and 1.25 mM ADP-Mg$^{2+}$ at 37 °C to measure maximal orotate production capacity. For the measurement of orotate production during CI-linked respiration, 1 mM malate, 5 mM pyruvate, and 5 mM glutamate were included. After 30 min incubation, the samples were placed on ice and mitochondria pelleted by

centrifugation (5 min 18,000 × *g* at +4 °C). Orotate in the supernatant was measured with a fluorometric method according to Yin et al.[109], the only modification being the increased K$_2$CO$_3$ concentration to compensate the buffering capacity of Mir05. The following components (100 μl each) were added to 100 μl supernatant: 4 mM 4-TFMBAO (4-trifluoromethylbenzamidoxime), 8 mM K$_3$[Fe(CN)$_6$], and 200 mM K$_2$CO$_3$. The mixture was incubated 4.5 min at 80 °C and cooled down on ice. The fluorescent product was determined with 350 nm excitation and 460 nm emission epifluorescence. Orotate standards were prepared in the same buffer as the sample were in (including all substrates required for mitochondrial respiration). The DHODH inhibitor teriflunomide (2 μM) decreased the measured orotate production by more than 90%.

## Citrate synthase activity

Citrate synthase activity in mitochondrial suspensions was measured by following 412 nm absorbance change upon the formation of 5,5′-dithiobis(2-nitrobenzoic acid) (DTNB) adduct with coenzyme A in a reaction mixture comprising 0.5 mM oxaloacetic acid, 300 μM acetyl-CoA, 100 μM DTNB, 0.1% Triton X-100, and 100 mM Tris-Cl pH 8.

## Enzymatic assays for ATP, NAD(H), and NADP(H)

Snap-frozen liver samples (-10 mg) were homogenized in buffer-phenol-choloroform-isoamyl alcohol (48:48:24:1) emulsion. The buffer phase comprised 25 mM Tris-Cl, 0.1 mM EDTA, pH 7.75 (at 0 °C). The aqueous phase was collected after centrifugation (6 min 15,000 × *g* at +4 °C) and washed twice with 1 ml diethyl ether to remove residual phenol. For NAD$^+$ and NADP$^+$ measurements, an aliquot was immediately stabilized by the addition of HCl to final concentration of 100 mM. Another aliquot was stabilized with 100 mM NaOH to prevent degradation of NADH and NADPH. A third aliquot was diluted 1:1000 with 25 mM Tris-acetate pH 7.75 and ATP measured using luciferin-luciferase reagent (Invitrogen, #A22066). Throughout the extraction, the samples were kept ice cold.

NAD$^+$ and NADH were measured using an assay based on cyclic amplification of MTT (methylthiazolyldiphenyl-tetrazolium bromide, 0.42 mM) reduction in the presence of phenazine ethosulfate (PES, 1.7 mM), *Saccharomyces cerevisiae* alcohol dehydrogenase (10 U/ml) and EtOH (10%). For measurement of NADP$^+$ and NADPH, glucose 6-phosphate dehydrogenase (1.4 U/ml) and glucose 6-phosphate (2.5 mM) served as the NADP$^+$ recycling enzyme-substrate pair. The assays were buffered to pH 8 with 60 mM Tricine-NaOH with 1 mM EDTA. The reduced and oxidized forms of the pyridine dinucleotides were separated utilizing their distinct pH lability (boiling for min 5 min 100 mM NaOH or HCl, respectively). The analyte concentrations were determined from the rate of 570 nm absorbance change against that of standards.

## Measurement of dNTPs

dNTPs were measured using a published method based on DNA polymerase, EvaGreen dye, and 197-nt single-stranded DNA templates[110]. Compared to the published protocol, a different oligonucleotide template for dGTP detection was used here (Supplementary Table 2).

## Liquid chromatography mass spectrometry-based metabolite analyses

Succinate, fumarate, and aspartate data derive from a targeted ultra-performance liquid chromatography mass spectrometry (UPLC-MS)-based protocol covering 16 tricarboxylic acid cycle-related metabolites[111]. The data on serine, glycine, dimethylglycine, methionine, threonine, and folate came from a targeted 101-metabolite UPLC-MS-based metabolomics[61].

Orotate, dihydroorotate, and N-carbamoyl-aspartate were extracted with 0.1% formic acid in acetonitrile with tissue disruption

using ball mill, freeze-thaw cycle in liquid nitrogen, and sonication. Chromatographic separation took place in Waters BEH Amide column (100 × 2.1 mm, ø 1.7 μm) at 35 °C with a flow rate of 0.4 ml/min. The elution solvents were 0.1% formic acid in water (A), and 0.1% formic acid in acetonitrile (B). The target metabolites were analyzed with UPLC- 6500 + QTRAP/MS (ABSciex) in negative ion mode, with multiple reaction monitoring (MRM) method: transition 155 → 111 for orotate, 157 → 113 for dihydroorotate, and 175 → 132 for N-carbamoyl-aspartate.

For measurement of rNTPs, polar metabolites were extracted with water-methanol-chloroform extraction. The methanolic extracts were dried in miVac Centrifugal Vacuum Concentrator. The chromatographic separation was performed in Waters Premier BEH C18 AX column (150 × 2.1 mm, ø 1.7 μm) at 40 °C. Elution solvents were 10 mM ammonium acetate in water, pH 9.0 (A) and acetonitrile (B) with the flow rate of 0.3 ml/min. Gradient elution started with 98% A, linearly decreased to 30% in 3 min, then to 10% in 1 min, back to 98% in 4.01 min, and stabilized for 1 min, with a total runtime of 5 min. Injection volumes were 1 μl. Adenosine triphosphate (ATP), cytidine triphosphate (CTP), guanosine triphosphate (GTP), and uridine triphosphate (UTP) were analyzed with UPLC-6500 + QTRAP/MS (ABSciex) in negative ion mode. Retention times for ATP, CTP, GTP, and UTP were 1.9, 1.15, 1.30, and 1.2 min, respectively. Each analyte was analyzed with the Multiple Reaction Monitoring (MRM) method with two transitions, quantitative and qualitative: ATP 506 → 159 (quan.), 506 → 408 (qual.); CTP 482 → 384 (quan.), 482 → 159 (qual.); GTP 522 → 159 (quan.), 522 → 424 (qual.); UTP 483 → 159 (quan.), 483 → 385 (qual.).

### Statistics

Throughout the figures the error bars present mean and 95% confidence interval of the mean, except in box plots and unless otherwise stated. The box blots follow the standard structure presenting the median and the quartiles (whiskers min. and max.). The differences between groups were tested with Welch's $t$-test (in the case of two groups), or one-way ANOVA followed by the indicated selected pairwise comparisons (Welch's $t$-statistics). The normality assumption of the tests was assessed by inspecting the histogram of residuals and by Shapiro–Wilk method. Where more appropriate, Kruskal–Wallis test and Mann–Whitney $U$ tests were used instead. $\chi^2$ test was performed on binary data. General linear model was applied for analyses involving additional variables (e.g. sex and body weight). All pairwise comparisons were performed using two-sided tests. Survival data were analyzed with log-rank test (Mantel–Cox). The experimental unit ($n$) in this study was one distinct animal. GraphPad Prism was utilized to analyze the survival data. IBM SPSS statistics was used for the other statistical tests.

### Reporting summary

Further information on research design is available in the Nature Portfolio Reporting Summary linked to this article.

## Data availability

The numerical data, and uncropped representative Western blots and the related blots used for quantifications are available as Source Data file accompanying this paper. The following public datasets available from ArrayExpress database were utilized in this study: E-MTAB-7416 [26] and E-MEXP-1503 [40]. Source data are provided with this paper.

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

## Acknowledgements

We thank Elisa Alppila for technical assistance, Jayasimman Rajendran for discussions and for help with DEXA, Eduard Daura Sarroca for discussions on cellular senescence and for suggesting the assessment of histone H3 cleavage, and Leah Biggs for guidance on terminology regarding the skin histology. We also thank the core facilities of University of Helsinki, the FIMM metabolomics unit (supported by HiLIFE and Biocenter Finland) for the liver metabolomics analyses, the Tissue Preparation and Histochemistry Unit (Department of Anatomy) and the Finnish Centre for Laboratory Animal Pathology (Faculty of Veterinary Medicine) for processing of histological samples, the Laboratory Animal Center of the University of Helsinki for the animal husbandry, and the AAV Gene Transfer and Cell Therapy Core Facility for the rAAV production. We acknowledge funding from Samfundet Folkhälsan, Jane and Aatos Erkko Foundation, the Foundation for Pediatric Research, Finska Läkaresällskapet, Medicinska Understödsföreningen Liv och Hälsa rf, Magnus Ehrnrooth Foundation, Alfred Kordelin Foundation, and Biomedicum Helsinki Foundation. Graphical illustrations in Fig. 10a, j were created with BioRender (www.biorender.com).

## Author contributions

J.P., V.F., and J.K. designed the study. J.P. wrote the first manuscript draft and prepared the figure panels. J.P., R.B., and J.K. performed the animal experiments and sample collection. R.B. and J.K. performed the cell culture experiments. M.M. was responsible for the electron microscopy analyses. N.S. developed and performed the UPLC-MS based quantification of DHODH-linked metabolites and rNTPs. J.P., R.B. and J.K. analyzed the histological data with J.P. being responsible for the automated image quantifications. The contributions to other methods were following: body composition analyses (J.P. and R.B.), SDS-PAGE and Western blot analyses (J.P., R.B., V.W. and J.K.), blue-native PAGE (J.P.), qPCR (J.P. and V.W.), transcriptomics analyses (J.P.), histochemistry and immunohistochemistry (J.K., J.P.), respirometry (J.P. and V.W.), mitochondrial ATP production (J.P. and R.B.), mitochondrial membrane potential (J.P.), enzymatic assays for ATP, NAD(H), NADP(H), and dNTPs (J.P.), mitochondrial $H_2O_2$ production (J.P.), DNA and RNA analyses (J.P. and V.W.), mitochondrial orotate production (J.P. and J.K.), rAAV cloning (J.K. and R.B.). J.P. was responsible for the statistics. All authors critically read and commented the manuscript, and J.P. and J.K. revised it accordingly.

## Competing interests

The authors declare no competing interests.
