## [Peer Review File · Nature Communications]

Mitochondrial complex III deficiency drives c-MYC overexpression and illicit cell cycle entry leading to senescence and segmental progeriaEditorial Note: This manuscript has been previously reviewed at another journal that is not operating a transparent peer review scheme. This document only contains reviewer comments and rebuttal letters for versions considered at *Nature Communications*.

REVIEWER COMMENTS

Reviewer #1 (Remarks to the Author):

The authors have made a serious effort to address all my comments. They now added new experimental data and made modifications in the text and in some figures. Their data clearly verify the concept, that complex III deficiency leads to senescence and premature aging. Accordingly, the revised manuscript is satisfactory I have no further comments. Notably, this article will also be an important contribution to our understanding of how mitochondrial dysfunction is linked to aging in general.

Reviewer #2 (Remarks to the Author):

The authors did a great job of responding!

Reviewer #4 (Remarks to the Author):

In this work, Purhonen and colleagues investigate the progeroid phenotype of *Bcs1^{lp.S78G};mt-Cybp.D254N* mice, a model of GRACILE syndrome. The authors show that this mouse model has several signs of premature aging in the tissues, which are not directly related to the bioenergetic defect, to increased ROS production or to an imbalance of redox status. Intriguingly, they found that increased expression of c-MYC mediates a dysregulation of the cell cycle.

The paper is potentially interesting, but I have two major concerns. First, I am not convinced that the model used in this study can be anyhow related to aging. *Bcs1^{lp.S78G};mt-Cybp.D254N* mice are characterized by a defect in respiratory complex III, due to aberrant processing of the Rieskie protein. However, the physiological role of BCS1L is poorly understood and several lines of evidence in the literature suggest that it may have additional roles beside complex III maturation. Accordingly, the GRACILE mutation is rather peculiar in its manifestations compared to other mutations in the same gene. The results of the experiments with AOX are puzzling. The fact that AOX was unable to correct the metabolic alterations, but could still ameliorate the phenotype, remains unclear and may well be due to other roles of BCS1L.

Second, no mechanistic explanation is given for c-MYC activation in the presence of the GRACILE mutation, and it is also unclear if this is specific to defects in complex III, including other mutations in BCS1L.

1 **Submission NCOMMS-22-42337-A**

2

3 **Referees' comments:**

4

5 **Reviewer #4 (Remarks to the Author):**

6 In this work, Purhonen and colleagues investigate the progeroid phenotype of *Bcs1^lp.S78G;mt-*
7 *Cybp.D254N* mice, a model of GRACILE syndrome. The authors show that this mouse model has
8 several signs of premature aging in the tissues, which are not directly related to the bioenergetic
9 defect, to increased ROS production or to an imbalance of redox status. Intriguingly, they found
10 that increased expression of c-MYC mediates a dysregulation of the cell cycle.

11 The paper is potentially interesting, but I have two major concerns. First, I am not convinced that
12 the model used in this study can be anyhow related to aging.

13 As for premature (pathological) aging, one theoretical rationale for this study was the progeroid
14 phenotype of the mtDNA replicative polymerase γ mutant (*Polg^{p.D257A}*) mice ("mutator mice"),
15 published by two groups nearly 20 years ago, and the discrepancy regarding the mechanisms of
16 the apparent premature aging in these mice. Accumulation of mtDNA mutations eventually
17 unavoidably leads to OXPHOS deficiency, yet there are few if any studies addressing how the
18 OXPHOS deficiency develops and leads to premature aging in the mutator mice. It is highly likely
19 that the progeroid phenotype of the mutator mice, which slowly develop CIII deficiency as part of
20 broader OXPHOS deficiency, and the striking juvenile progeroid phenotype of the *Bcs1^lp.S78G* mice,
21 which become CIII deficient very rapidly after weaning, have similar underlying mechanisms.

22 As for aging in general, there are differing opinions about how similar pathological premature
23 aging, such as in progerias, is to normal aging. Nevertheless, they share several features and
24 markers, as shown by extensive literature. As for the normal aging, the modified Harman's free
25 radical theory of aging, positing that oxidative damage to mtDNA is critical to aging, has never
26 been truly proved or disproved in a physiologically relevant mammalian model. Here, we show
27 that increased mitochondrial reactive oxygen species production is dispensable for accelerated
28 aging of mitochondrial origin in this *in vivo* model – a very important conclusion on its own right.
29 Secondly, secondary deterioration of mitochondria is relatively common in aging-associated
30 diseases. Thirdly and perhaps most importantly, accumulation of senescent cells is widely
31 accepted to accompany aging. Therefore, our findings in this model are related to both premature
32 and normal aging in several ways.

33 Kujoth, G. C. et al. Mitochondrial DNA mutations, oxidative stress, and apoptosis in mammalian
34 aging. *Science* 309, 481–484 (2005).

35 Trifunovic, A. et al. Premature ageing in mice expressing defective mitochondrial DNA polymerase.
36 *Nature* 429, 417–423 (2004).

37 *Bcs1^lp.S78G;mt-Cybp.D254N* mice are characterized by a defect in respiratory complex III, due to
38 aberrant processing of the Rieskie protein. However, the physiological role of BCS1L is poorly
39 understood...

40 We are very surprised about this claim because the physiological function of BCS1L is likely
41 understood better than that of any other respiratory chain assembly factor (see references
42 below). Already in the early 1990s, studies in yeast showed that *Bcs1* was required for
43 RISP/UQCRFS1 assembly. Since then, several high-profile papers have shown similar function in

44 mammalian mitochondria, and elaborated the mechanism of function, confirming that BCS1L is
45 specific assembly factor, or rather a translocase, for the RISP subunit of CIII. These studies
46 culminated in the publication of cryoelectron microscopy structures of both the yeast and mouse
47 BCS1L membrane complex. These impressive structural studies delineated a mechanism for the
48 translocation of RISP and its bulky fully folded iron-sulfur domain across the mitochondrial inner
49 membrane without compromising the electrochemical gradient. This very delicate translocation
50 mechanism very likely means that there are no other substrates for BCS1L than RISP.

51 Nobrega et al. BCS1, a novel gene required for the expression of functional Rieske iron-sulfur
52 protein in *Saccharomyces cerevisiae*. EMBO J. 1992 Nov;11(11):3821-9.

53 Fölsch et al. Internal targeting signal of the BCS1 protein: a novel mechanism of import into
54 mitochondria. EMBO J. 1996 Feb 1;15(3):479-87.

55 Kotarsky et al. Characterization of complex III deficiency and liver dysfunction in GRACILE
56 syndrome caused by a BCS1L mutation. Mitochondrion. 2010 Aug;10(5):497-509.

57 Wagener et al. A pathway of protein translocation in mitochondria mediated by the AAA-ATPase
58 Bcs1. Mol Cell. 2011 Oct 21;44(2):191-202.

59 Ostojić et al. The energetic state of mitochondria modulates complex III biogenesis through the
60 ATP-dependent activity of Bcs1. Cell Metab. 2013 Oct 1;18(4):567-77.

61 Tang et al. Structures of AAA protein translocase Bcs1 suggest translocation mechanism of a
62 folded protein. Nat Struct Mol Biol. 2020 Feb;27(2):202-209.

63 Kater et al. Structure of the Bcs1 AAA-ATPase suggests an airlock-like translocation mechanism for
64 folded proteins. Nat Struct Mol Biol. 2020 Feb;27(2):142-149.

65 ...and several lines of evidence in the literature suggest that it may have additional roles beside
66 complex III maturation.

67 Without references or other elaboration by the referee it is difficult to know what additional roles
68 the referee hints at. During the 20 years we have studied BCS1L, we have never come across
69 literature showing evidence for BCS1L translocating any other protein than RISP across the
70 mitochondrial inner membrane. Neither are we aware of studies showing localization of BCS1L to
71 other organelles than mitochondria. In the *Bcs1^{p.S78G}* mice, CIII activity decreases linearly after
72 weaning to as low as <20% of WT and this correlates with the loss of RISP from CIII and disease
73 progression, as shown in our several publications previously. Of course, it is impossible to rule out
74 potential as yet unknown functions of BCS1L.

75 Accordingly, the GRACILE mutation is rather peculiar in its manifestations compared to other
76 mutations in the same gene.

77 The phenotypic spectrum of BCS1L-related mitochondrial diseases was reviewed recently (Hikmat
78 et al. 2021). In this review, Table 3 summarizes the phenotypes of patient carrying the Ser78Gly
79 mutation, Ser78Gly compound heterozygotes and other BCS1L mutations and shows that GRACILE
80 syndrome is not peculiar compared to the other phenotypes. It's more severe, with earliest onset
81 and shortest survival, but not qualitatively that different.

82 Hikmat et al. Expanding the phenotypic spectrum of BCS1L-related mitochondrial disease. Ann Clin
83 Transl Neurol. 2021 Nov;8(11):2155-2165. doi: 10.1002/acn3.51470.

84 The results of the experiments with AOX are puzzling. The fact that AOX was unable to correct the
85 metabolic alterations, but could still ameliorate the phenotype, remains unclear and may well be
86 due to other roles of BCS1L.

87 The functions of CIII are electron transfer to cytochrome *c* coupled to proton translocation to
88 generate proton motive force, reoxidation of coenzyme Q (CoQ), and production of ROS. In
89 theory, AOX can rescue the CoQ oxidation but not proton translocation nor CIII-dependent ROS-
90 signaling. Therefore, the lack of effect of AOX on OXPHOS is in line with literature and theory. We
91 find it very unlikely that AOX would affect some function other than CoQ oxidation, even if the
92 BCS1L mutation would cause some secondary CIII-independent effects. The fact that AOX
93 suppressed the c-MYC induction, cellular proliferation, and ameliorated the phenotype but did not
94 correct the metabolic alterations related to the OXPHOS deficiency is one of the main points of our
95 study. In brief, the AOX experiment demonstrates that non-replicating parenchymal cells can
96 tolerate very severe OXPHOS deficiency, whereas proliferation of these cells against the OXPHOS
97 deficiency leads to a catastrophic outcome, ultimately segmental progeria (Discussion, lines 415-
98 421).

99 Second, no mechanistic explanation is given for c-MYC activation in the presence of the GRACILE
100 mutation,

101 Pursuit of the potential signals leading to the c-MYC upregulation was out of the scope of this
102 already extensive paper, but is an important question and a subject of our ongoing work. There
103 are several possibilities: 1) c-MYC is induced by metabolic cues related to a compensatory
104 Warburg-like metabolic state similar to the mode of c-MYC overexpression in some cancers. 2) c-
105 MYC is induced by stress-responsive growth factors such as the EGFR ligand amphiregulin (AREG)
106 that is one of the most highly upregulated genes in the *Bcs1l* mutant mice (Supplementary Fig. 1i)
107 and also a recently identified mitokine (Hino et al. 2022). 3) c-MYC is a mitochondrial retrograde
108 signaling factor similar to the yeast homologue Rtg1. We have added some discussion on these
109 possibilities (Discussion, lines 484-497).

110 Hino, Y. et al. Mitochondrial stress induces AREG expression and epigenomic remodeling through
111 c-JUN and YAP-mediated enhancer activation. *Nucleic Acids Research*. gkac735 (2022)
112 doi:10.1093/nar/gkac735.

113 and it is also unclear if this is specific to defects in complex III, including other mutations in BCS1L.

114 This is an important point. As we explain in the Results and the Discussion (lines 329-330 and 484-
115 490) the c-MYC upregulation is not unique to the GRACILE mutation as there are evidence of c-
116 MYC upregulation in other models of OXPHOS deficiency. The main aim of this study was to be a
117 proof-of-concept that isolated CIII deficiency is sufficient to cause segmental progeria. Premature
118 aging-like manifestations remain poorly reported or unrecognized in OXPHOS diseases. Examples
119 of possible OXPHOS-linked progerias, however, do exist such as due to mutations in SLC25A24,
120 MTX2, and very recently in TOMM7 as stated in the Discussion (lines 429-439). How OXPHOS is
121 affected in these extremely rare diseases remains poorly understood. In summary, we believe that
122 an OXPHOS defect exceeding a certain threshold is critical to the progeroid pathology. In patients
123 carrying other BCS1L mutations there may be segmental manifestations depending on the
124 affected tissue and level of CIII activity loss.

REVIEWERS' COMMENTS

Reviewer #4 (Remarks to the Author):

The authors made an effort to reply to my concerns.